# HOW DO LANGUAGE MODELS COMPOSE FUNCTIONS?

## ABSTRACT

While large language models (LLMs) appear to be increasingly capable of solving compositional tasks, it is an open question whether they do so using compositional mechanisms. In this work, we investigate how feedforward LLMs solve two-hop factual recall tasks, which can be expressed compositionally as $g(f(x))$. We first confirm that modern LLMs continue to suffer from the "compositionality gap": i.e. their ability to compute both $z = f(x)$ and $y = g(z)$ does not entail their ability to compute the composition $y = g(f(x))$. Then, using logit lens on their residual stream activations, we identify two processing mechanisms, one which solves tasks *compositionally*, computing $f(x)$ along the way to computing $g(f(x))$, and one which solves them *directly*, without any detectable signature of the intermediate variable $f(x)$. Finally, we find that which mechanism is employed appears to be related to the embedding space geometry, with the idiomatic mechanism being dominant in cases where there exists a linear mapping from $x$ to $g(f(x))$ in the embedding spaces.

## 1 INTRODUCTION

*Compositional behavior* (McCurdy et al., 2024) is widely considered essential for flexible and general intelligence (Szabó, 2024). A long-running debate has asked whether compositional *behavior* necessarily entails compositional *representations* and *processes*. One the one hand, formal languages based on compositional syntax and semantics are guaranteed to support certain types of invariance and generalization, making them compelling models for how humans might achieve abstract cognitive abilities like language and logic (Fodor, 1975; Quilty-Dunn et al., 2023). On the other hand, critics are quick to point out that humans frequently deviate from ideal compositional and logical behavior, suggesting that some other mechanism must underlie our advanced cognition (Kahneman & Tversky, 1972; Evans, 2002).

Large language models (LLMs) provide an opportunity to revisit this debate in a new light. LLMs exhibit behavior that is at least ostensibly compositional, and which is not easily explained away by trivially non-compositional mechanisms (McCoy et al., 2023; Griffiths et al., 2025). However, LLMs also lack the kinds of explicit symbolic architectural components that have long been assumed necessary for such compositionality. This provides an opportunity to ask: do LLMs produce compositional behavior by invoking compositional processes, or do they rely on something more idiomatic instead?

We offer an initial investigation into this question, focusing on a set of two-hop factual retrieval tasks, such as: given a book's title, output that book's author's birth year. All of the tasks we consider can be formally expressed as $y = g(f(x))$ and are thus defensibly "compositional" in the sense invoked in traditional symbolic models. We are interested in whether LLMs solve such tasks by approximating the mapping from $x$ to $y$ *compositionally*, by computing the intermediate variable $z = f(x)$, or *directly*, without a readily-detectable representation of any such $z$. We find that:

1. Models' ability to compute both $x \to f(x)$ and $f(x) \to g(f(x))$ does not entail their ability to compute $x \to g(f(x))$. This extends earlier findings on the "compositionality gap" (Press et al., 2022), showing that the gap holds for modern models and on a larger set of tasks. This gap is not trivially reduced in larger models or even necessarily by reasoning models (Sec. 3).

2. Models exhibit both *compositional* processing mechanisms and *direct* processing mechanisms, as defined above. The type of mechanism is only weakly associated with accuracy, suggesting that LLMs are able to use both effectively to compute correct answers (Sec. 4).

3. The choice of mechanism is mediated by the geometry of the input embedding space. Specifically, when there exists a linear mapping from $x$ in the input embedding space to $g(f(x))$ in the output unembedding space, the LLM tends to favor direct computation over compositional processing (Sec. 5).

## 2 TASK SETUP

Our tasks involve solving a composition $g(f(x))$ from an input $x$, using in-context learning (ICL) and where $f$ and $g$ are some pre-defined functions. See Table 1 for the full list of tasks we use. We choose common functions $f$ and $g$ that models might learn through their pre-training and for which the inputs and outputs are lexical units. This enables us to use well-established tools for analyzing the mechanisms and latent computations in Transformer models, focusing on a few token positions (i.e. residual streams) and a single autoregressive forward pass.

We design the set of tasks in our investigation to cover a qualitative variety of functions, such as arithmetic, factual recall, lexical functions, translation, rotation, and string manipulation. By construction, all of our tasks can be computed by applying $f$ and then $g$, yielding the causal hops $x \rightarrow f(x) \rightarrow g(f(x))$. Some tasks (e.g. commutative tasks) can also be computed through the hops $x \rightarrow g(x) \rightarrow g(f(x))$ — in which case, the intermediate $z$ may also equal $g(x)$. We differentiate these further, and also describe our dataset construction methodologies (including our sources and pre-processing), in Appendix A.

In our experiments, we randomly sample 10 in-context examples for a given task and query. Each in-context example is formatted with a "Q: {input} \n A: {output} \n\n" prompting structure and the test query is formatted with "Q: {input} \n A:".

**Limitations** Our experimental design primarily focuses on autoregressive language models permitted one token for generation (rather than e.g. reasoning models) and on mechanisms that are discoverable using current widely-accepted interpretability methods. There are certainly many interesting compositional and non-compositional mechanisms that are employed by LLMs which are not in the scope of the present study. The mechanisms we describe here are part of the larger story and thus warrant study, but we do not intend to imply that such mechanisms are the whole story of how LLMs process complex tasks.

Table 1: List of our tasks. The compositional function $(g \circ f)$ is constructed by $f$ and $g$ here. We list the number of examples (#) in each task's dataset, along with the variables $x$, $f(x)$, and $g(f(x))$ for one random example. We list $g(x)$ and $f(g(x))$ for tasks that define them in Appendix A.

| $f$ | $g$ | # | $x \rightarrow f(x) \rightarrow g(f(x))$ |
|---|---|---|---|
| Word → Antonym | English → Spanish | 2398 | bogus → authentic → auténtico |
| Word → Antonym | English → German | 2398 | philosophical → practical → praktisch |
| Word → Antonym | English → French | 2398 | excessive → insufficient → insuffisant |
| Book → Author | Author → Birth Year | 2228 | The Boy in the Striped Pyjamas → John Boyne → 1971 |
| Song → Artist | Artist → Birth Year | 958 | Heartbreak Hotel → Elvis Presley → 1935 |
| Landmark → Country | Country → Capital | 1385 | Taq-i Kisra → Iraq → Baghdad |
| Park → Country | Country → Capital | 743 | Mount Rainier National Park → United States → Washington, D.C. |
| Movie → Director | Director → Birth Year | 2180 | Cape Fear → Martin Scorsese → 1942 |
| Person → University | University → Year | 4992 | Andi Gutmans → Technion – Israel Institute of Technology → 1924 |
| Person → University | University → Founder | 4996 | Ezra Abbot → Bowdoin College → James Bowdoin |
| Product → Company | Company → CEO | 1904 | NES-101 → Nintendo → Shuntaro Furukawa |
| Product → Company | Company → HQ | 2276 | Toyota Alphard → Toyota → Toyota |
| x + 10 | 2x | 1000 | 699 → 709 → 1418 |
| x + 100 | 2x | 1000 | 922 → 1022 → 2044 |
| x mod 20 | 2x | 1000 | 891 → 11 → 22 |
| Word → Numeric | 2x | 1000 | one hundred and forty-eight → 148 → 296 |
| Word[:-1] | Word[::-1] | 2946 | responsible → responsibl → lbisnopser |
| Rotate(RGB, 120°) | RGB → Name | 1000 | 8a735a → 598a73 → dimgray |

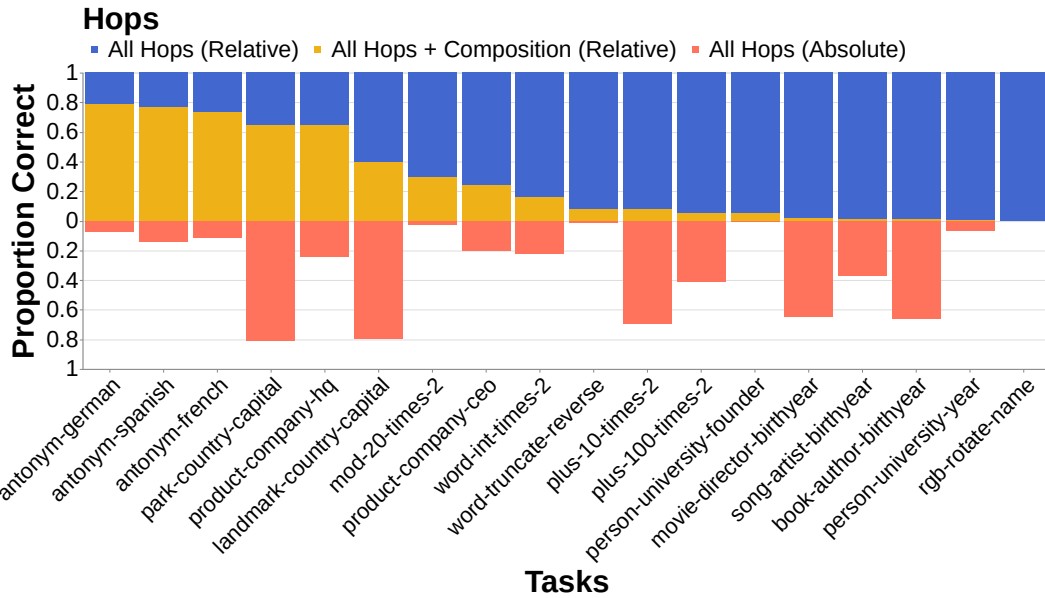

Figure 1: Compositionality gap for Llama 3 (3B) on our tasks. Red bar represents examples for which the model is able to solve all causal hops, out of all examples (absolute). Blue and yellow bars are relative to the red bar: they show proportions of examples out of those in the red bar. Blue represents the same examples as red and yellow represents those for which the model is able to additionally solve the composition. Correlation between red and yellow bars is $r^2 = 0.00$.

## 3 COMPOSITIONALITY GAP

Press et al. (2022) documented a "compositionality gap" in LLMs, showing that they consistently fail to solve compositions, despite solving the hops independently. Press et al. (2022) tested the GPT-3 family of models with natural language questions about celebrities and encyclopedic knowledge that required two-hops of factual recall. We confirm and extend this finding by testing modern LLMs on a larger set of compositional tasks.

### 3.1 EXPERIMENTAL DESIGN

We prompt models with input → output mappings between lexical units.[1] We measure models' predictive accuracies using the ICL prompts from Sec. 2, greedy sampling, and the exact match evaluation metric. The *compositionality gap* is defined as the proportion of examples for which a model answers both $x \rightarrow f(x)$ and $f(x) \rightarrow g(f(x))$ correctly,[2] but $x \rightarrow g(f(x))$ incorrectly.

We test the Llama 3 (3B) model on all of our tasks, using all available examples. We also test a wider set of models (including those from Llama 3, OLMo 2, DeepSeek, and GPT model families) on 4 tasks: antonym-spanish, plus-100-times-2, park-country-capital, and book-author-birthyear (which capture a representative set of processing signatures from Sec. 4). We aggregate metrics over these tasks and use 100 examples per task for testing.

### 3.2 RESULTS

We show performance of the Llama 3 (3B) model on our tasks in Fig. 1. We clearly find a compositionality gap: the model is unable to solve the composition in 20–100% (varying by task) of examples for which it can solve all hops. We show the performances of our other models in Fig. 2.

---

[1]Note that this represents a methodological difference from Press et al. (2022), who prompted with long-form questions. Our format is chosen to fit with the interpretability methods we use in later sections.

[2]We extend this definition to further require success at $x \rightarrow g(x)$ and $g(x) \rightarrow g(f(x))$ in tasks where these are valid hops.

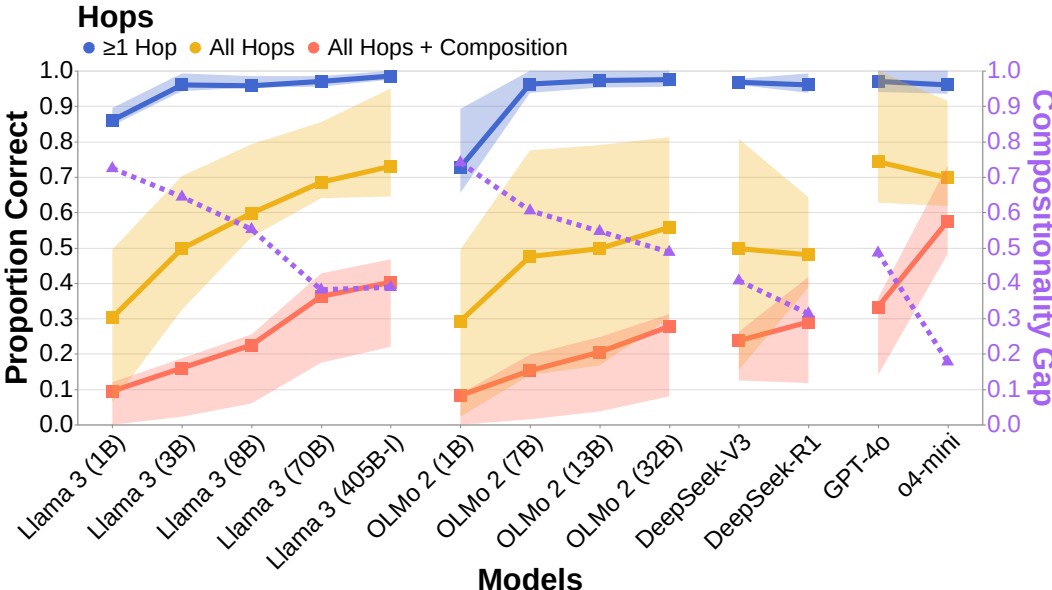

Figure 2: Compositionality gap (dashed purple line; lower is better) of various models aggregated over 4 tasks (100 examples each). Blue, yellow, and red lines show proportions of examples for which models correctly solve combinations of hops and the composition. Purple line shows the relative gap between yellow and red: the proportion of examples for which the model cannot solve the composition, out of those for which it can solve all hops. "-I" indicates the instruction-tuned variant of Llama 3 (405B). Error bands show interquartile range.

We find the compositionality gap does reduce with size from 72% → 39% (Llama 3, 1B → 405B) and 74% → 49% (OLMo 2, 1B → 32B). However, the gap clearly remains and we find monotonically diminishing improvements for both model families with respect to size. We plot the gap against model parameters and layers in Appendix C. In fact, the gap shows no improvement at all between the 70B and (instruction-tuned) 405B parameter Llama 3 models.

We also compare reasoning models (o4-mini and DeepSeek-R1; allotted a budget of 2000 reasoning tokens) against same-generation, non-reasoning models (GPT-4o and DeepSeek-V3) in Fig. 2. We find some reduction (41% → 31%) in the compositionality gap in the case of DeepSeek's reasoning model and significant reduction (49% → 18%) in the case of o4-mini. As o4-mini is proprietary (and both this and GPT-4o have additional "external tool-use" capabilities), it is difficult to speculate about the exact causes for these improvements. However, it is notable that even with advanced reasoning models, the gap does not necessarily disappear entirely.

## 4 ANALYZING PROCESSING MECHANISMS

We next try to understand *how* the model correctly computes compositions in cases where it is successful. Our intuition is based on prior work from Merullo et al. (2024) which identifies a processing signature in models that solve one-hop relational tasks. That work shows that models predicting $y = f(x)$ iteratively surface vocabulary representations — first for $x$ and then for $y$ — in the residual stream. This "crossover" point was interpreted as evidence of the function $f$ being applied to the argument $x$ in order to yield the final answer $f(x)$ and was localized to specific computations in the MLPs.

In this section, we ask whether an analogous signature will emerge in the case of compositional functions, $g(f(x))$. That is, can we find distinct intermediate representations for $x$, followed by $f(x)$, and then $g(f(x))$ during the model's processing?

Here, we employ analyses most similar to Biran et al. (2024) and Yang et al. (2025) in the context of our evaluation (see Sec. 7 for further discussion on these works). We also join other works in

identifying stages of processing within language models (Tenney et al., 2019; Merullo et al., 2024; Lepori et al., 2024).

## 4.1 EXPERIMENTAL DESIGN

We rely on existing methods which allow us to analyze processing signatures that are interpretable using the vocabulary space of the model (nostalgebraist, 2020; Geva et al., 2022). We specifically use *logit lens* (nostalgebraist, 2020), a method which projects intermediate representations into the vocabulary space using the language modeling head. We also include results in Appendix F using the token identity patchscope (Ghandeharioun et al., 2024) as an alternative decoding method to logit lens. We find that both methods yield similar findings.

We follow the approach from Merullo et al. (2024) to identify the processing signature of models that solve our compositional tasks and, in particular, representations of the intermediate variables, $f(x)$ and $g(x)$, prior to those for $g(f(x))$. We specifically use logit lens to analyze the residual streams corresponding to the computation $x \rightarrow g(f(x))$ and measure the reciprocal rank of our variables at each layer (see Appendix B for more details). We also use the maximum reciprocal rank of our intermediate variables across the layers as a heuristic for their overall presence in the computation.

We conduct this analysis with the Llama 3 (3B) model. We exclusively analyze examples where the model can solve all requisite hops. To ensure sufficient sample sizes, we exclude any task with fewer than 10 such examples where the model can also successfully solve the composition. In particular, these excluded tasks include `song-artist-birthyear`, `person-university-year`, `person-university-founder`, `mod-20-times-2`, `word-truncate-reverse`, and `rgb-rotate-name`. We show results for these tasks in Appendices D and E.

## 4.2 RESULTS

Fig. 3a shows the relative presence of each of the variables, across layers and aggregated over all instances in which the model ultimately produced the correct answer. In such cases, we see a very clear peak signal for the intermediate variable $f(x)$, as expected, between those for $x$ and $g(f(x))$. Interestingly, this signal is much less clear for cases in which the model ultimately produces the incorrect answer (Fig. 3b). However, upon further inspection, there is little evidence of a causal relationship here, which we discuss further in Appendix E.

There are also plenty of individual examples in which the model produces a correct answer without showing any signature of the intermediate variables, and there is only a weak correlation by task ($r^2 = 0.22$) between predictive accuracy (measured as in Sec. 3.1) and the presence of intermediate variables as measured by our heuristic (Sec. 4.1).

Figs. 3c to 3f show model processing signatures for a few tasks, aggregated over cases in which the model produces correct answers. We see, for example, that there is a clear signature in the `antonym-spanish` task (Fig. 3c) for the intermediate computation of $f(x)$, the word's antonym, before it is translated into Spanish. In contrast, for the `movie-director-birthyear` task (Fig. 3d), there is no decodable signal for $f(x)$, the movie's director, before the model produces their birth year. This variation can be seen in qualitatively similar tasks as well: tasks with the same basic arithmetic structure (Figs. 3e and 3f) only sometimes carries detectable signatures of $f(x)$ or $g(x)$, depending on the task's operand (e.g. 10 or 100). We show processing signatures for the remaining tasks in Appendix D and for all tasks, aggregated over unsuccessful cases, in Appendix E.

## 5 COMPOSITIONAL PROCESSING AND EMBEDDING SPACE LINEARITY

Given that there is significant variation in whether or not the LLM solves a task compositionally (i.e. how strongly they appear to compute the intermediate variables), we next ask why this variation occurs. It is well-known that embedding spaces can capture relational information in their geometry (Mikolov et al., 2013; Hewitt & Manning, 2019). Moreover, Hernandez et al. (2024) shows that some subject $\rightarrow$ object relations can be represented by a single linear transformation from a language model's residual stream activations to its unembedding space. Following this, we propose and test a

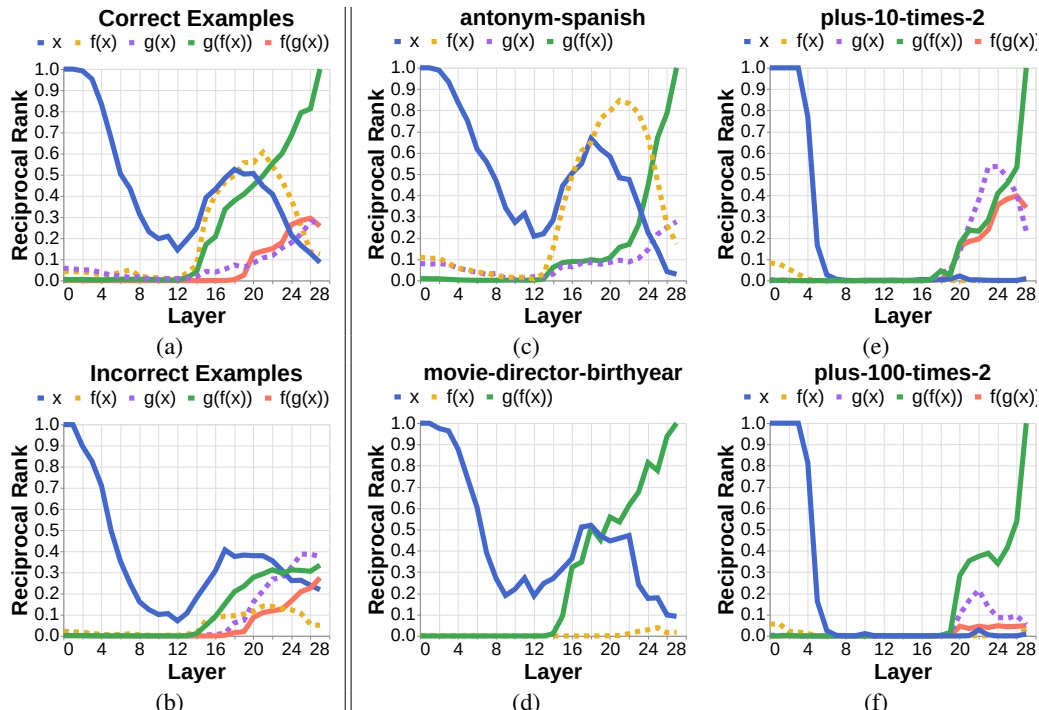

Figure 3: (a–b) Processing signatures aggregated over examples (across all tasks) in which Llama 3 (3B) solves all hops correctly, but the composition (a) correctly or (b) incorrectly. (c–f) Processing signatures for particular tasks — aggregated over examples where the model correctly solves all hops and the composition. (a–f) Lines show reciprocal ranks of relevant variables (decoded using logit lens) from residual streams corresponding to $x \rightarrow g(f(x))$. Intermediate variables are shown with dashed lines. The incorrect composition, $f(g(x))$, is shown by the red line when not distinct from $g(f(x))$.

hypothesis that language models could process compositional functions in one hop if they are directly represented as a linear transformation between the embedding and unembedding spaces.

## 5.1 EXPERIMENTAL DESIGN

To investigate our hypothesis, we fit a linear transformation for each task using least squares regression from $x$ (average embedding across tokens) to $g(f(x))$ (first token unembedding) on 100 examples.[3] We quantify the "linearity" of this transformation using its reconstruction accuracy (measured via cosine similarity) on the remaining examples. We quantify how "compositional" the processing is using our heuristic metric which captures the strength of the signal for the intermediate variables, $f(x)$ and $g(x)$ (see Sec. 4). We again restrict our analysis to examples where the model is successful on all hops and the composition, as well as tasks with at least 10 such examples.

## 5.2 RESULTS

Fig. 4a shows the that there is a a strong inverse correlation ($r^2 = 0.53$) between the linearity of the representation and the compositionality of the processing. That is, the more linear the representation of a relation is in the embedding spaces, the more likely the model is to display *idiomatic* (as opposed to *compositional*) processing.

This correlation is computed by averaging linearity and compositionality across instances for each task. Fig. 4b shows the de-aggregated distribution of our "compositionality" metric across the

---

[3]See Appendix H for additional analyses which consider correlations with the linearities of the individual hops (rather than the compositional task).

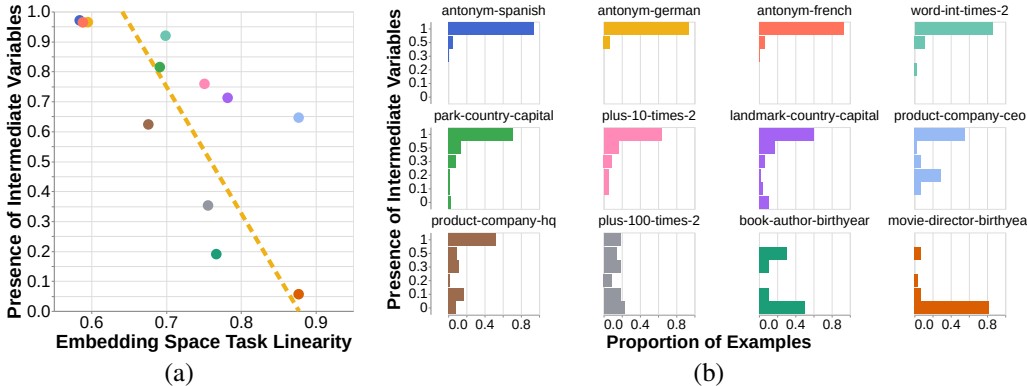

Figure 4: (a) Strong correlation across tasks between presence of intermediate variables (heuristic from Sec. 4.1 based on reciprocal rank; on average across examples) and embedding space linearity ($r^2 = 0.53$). Conversely, accuracy is weakly correlated with these intermediate variable ($r^2 = 0.22$) and linearity ($r^2 = 0.13$) metrics. (b) Distribution of examples for each task, shown as a histogram of intermediate variable reciprocal ranks. (a–b) Colors refer to corresponding tasks between points in (a) and histograms in (b).

examples. For some tasks, it appears that nearly all individual examples behave the same way. For example, nearly every instance of the `antonym-spanish` task displays a compositional processing signature, while almost every instance of the `movie-director-birthyear` task displays an idiomatic one. On the other hand, this distribution is more uniform for other tasks, such as `plus-100-times-2`. This distribution appears to be bimodal across all examples: 82% have very low ($< 0.1$) or high ($\geq 0.5$) values for compositionality.

## 6 DISCUSSION

**Summary of Findings**   Our results suggest that tasks which appear to have the same computational structure may nonetheless be processed differently by LLMs. In particular, we consider functions which appear compositional in a formal sense — i.e. they can be represented as $y = g(f(x))$ for some reasonably defined $f$ and $g$. We find evidence that LLMs only sometimes process such functions compositionally, showing evidence of representing or computing the value of $z = f(x)$ on the way to computing $y$. In other cases, LLMs appear to map $x$ to $y$ directly. Which of these processes is invoked appears to be related to how well the relationship between $x$ and $y$ is represented in the embedding space, e.g. as a result of pretraining (Merullo et al., 2025).

**Implications for Theories of Compositionality**   There is a long-running debate about the degree to which compositional behavior (McCurdy et al., 2024) requires compositionality at the level of mechanisms. The two sides of this debate have often talked past each other, often using different types of computational architectures in order to model different aspects of behavior, for example, using explicitly compositional symbolic systems to model formal domains (Lake et al., 2017; Ellis et al., 2023) and using distributional or neural systems to model humans' more idiomatic performance (Erk, 2012; Lampinen et al., 2024).

Attempts to find compromises or "hybrid" systems often consist of neuro-symbolic systems which are designed top-down (Andreas et al., 2016; Ellis et al., 2018). Large language models offer an alternative approach for advancing this debate. LLMs have proven capable of a range of behaviors that have traditionally required compositionality — e.g. generating language and writing formal computer code. However, LLMs lack the explicit symbolic mechanisms traditionally associated with such behaviors. Using methods from interpretability to understand how LLMs represent such functions internally enables us to approach the question in a "bottom up" manner, potentially offering novel hypotheses about the mechanisms that can generate behavior that is sometimes systematic and other times heuristic, as is the case in humans (Russin et al., 2025).

Our results suggest that LLMs employ a mix of compositional and idiomatic processing, and that the choice of mechanism is related to the representations of the functions that result from pretraining. This offers an interesting perspective on one question that is frequently at the heart of discussions of compositionality — i.e. what are the primitives and where do they come from Carey (2011)? The relationship between linearity in embedding space and compositionality of processing presented here suggests an attractive hypothesis that the primitives are those things which are well represented as a result of (pre-)training, and that compositional mechanisms are invoked to handle those things which are not sufficiently well represented. Future work in this direction would likely yield interesting new results and topics for debate.

**Relationship to work on compositional generalization**   The work presented here concerns the (apparent) compositionality of the processing mechanism, but does not directly relate this mechanism to an LLM's capacity for compositional generalization. The majority of work on compositionality in neural networks (and LLMs) concerns compositional generalization, and the compositionality researchers surveyed by McCurdy et al. (2024) overwhelmingly agree that existing language models are insufficient in this regard. This belief is supported by evidence from many prior works (Sec. 7) and our investigation in Sec. 3.

Our work suggests that models employ both compositional and direct mechanisms to solve tasks. Intuitively, we would expect there to be a relationship between the use of the mechanism and the ability to generalize — i.e. the compositional mechanism should support generalization better than the idiomatic mechanism ("memorization"). However, we do not test this intuition directly in this paper. Future work could do so by employing causal interventions on the intermediate variables, for example (see Appendix G for some initial investigations). This would likely present new complexities and challenges that would enrich our understanding of compositionality, and of the relationships between mechanisms and behaviors in LLMs in general.

# 7   RELATED WORK

**Latent multi-hop reasoning**   Our work is most closely related to recent or concurrent works which also study latent two-hop reasoning in large language models. Yang et al. (2024a) use causal interventions to identify the existence of the hops in the latent computation and whether they co-occur. Biran et al. (2024) employ the entity description patchscope (Ghandeharioun et al., 2024) to inspect intermediate representations and localize the hops, finding they are resolved in different layers and token positions. They propose a representational intervention ("backpatching") to correct failures based on this finding. Finally, Yang et al. (2025) use logit lens to analyze intermediate representations and consistently find a "compositional" processing signature across their tasks. Our work employs all of these interpretability methods (Sec. 4 and Appendices F and G) to analyze the hops, but specifically highlights and investigates the duality of the compositional vs. direct processing mechanisms. All works (including our own) test different sets of tasks, make experimental design decisions according to their independent goals,[4] and make findings in context of their own experiments.

Among other works in this domain, Wang et al. (2024) trains a language model on synthetic compositional data and identifies a multi-hop reasoning circuit in this model. Shalev et al. (2024) conduct a distributional analysis (considering semantic category spaces, rather than individual tokens) using logit lens. Li et al. (2024); Yu et al. (2025) also propose interventions on intermediate representations and mechanisms to solve failure cases. Yang et al. (2024b) conduct an evaluation that is intentionally designed to omit opportunities for models to exploit shortcuts.

**Compositionality**   Compositionality is long-studied (Fodor & Pylyshyn, 1988; Partee, 2004) but exact definitions evade general consensus. Russin et al. (2024) and McCurdy et al. (2024) offer recent overviews on the topic in the context of large language models. Russin et al. (2024) provide a historic account of compositionality and review studies of compositionality generalization in neural networks. McCurdy et al. (2024) survey compositionality researchers on how to define and evaluate compositional behavior in neural networks. These researchers agree that our current representational

---

[4]One notable example is that, while Yang et al. (2024a) and Biran et al. (2024) prompt their models with $f$ and $g$ (e.g. "The mother of the singer of {x} is {y}"), we omit this information from our prompts (i.e. more simply "Q: {x} \n A: {y}") to avoid inducing bias towards the compositional mechanism.

analyses are insufficient for evaluating models, but are divided about whether our behavioral analyses are sufficient.

In a partial effort towards defining compositionality, Hupkes et al. (2020) identify five particular aspects of compositionality and propose tests for each using a synthetic, fully compositional translation task. Systematicity is one such aspect and is prominently studied: see Vegner et al. (2025) for a survey of benchmarks for systematic generalization. Our work — in which we test whether $f(x)$ is evaluated before $g(f(x))$ — is closest to Hupkes et al. (2020)'s aspect of localism, in which "smaller constituents are evaluated before larger constituents".

Among many other works, Johnson et al. (2017), (Keysers et al., 2019), Lake & Baroni (2018), Hupkes et al. (2020), and Kim & Linzen (2020) offer prominent benchmarks that behaviorally test for compositional generalization in neural networks trained from scratch on compositional data. These works generally show that such models perform poorly on generalization, or at least poorly implement the compositional processes that underlie the data. Press et al. (2022) and Ma et al. (2023) continue to show significant failures in compositional generalization in pre-trained models. On the other hand, Furrer et al. (2020) points out that pre-training a masked language model rivals or outperforms architectures specifically designed for the SCAN (Lake & Baroni, 2018) and CFQ (Keysers et al., 2019) generalization benchmarks. Lepori et al. (2023) finds that neural networks learn to implement compositionality structurally in their weights, supporting this claim against the need for specialized symbolic mechanisms.

**Compositionality of functions**    Several works consider how language models solve compositions of functions (rather than specifically multi-hop reasoning tasks). Dziri et al. (2023) studies how language models autoregressively solve such tasks, like multi-digit multiplication, by inspecting their scratchpads. Wattenberg & Viégas (2024) propose mechanisms which neural networks could use to implement relational compositions. Yu et al. (2023); Todd et al. (2024) propose zero-shot methods to invoke compositions of functions in language models that have learned the primitive functions. Zhou et al. (2024) find that language models can compose functions with meta-learning in a way that imitates human behavior.

## LIMITATIONS

In this work, we primarily analyze the computation that occurs in a single forward pass of the Llama 3 (3B) model. It is also necessary to understand how other models (e.g. larger models, reasoning models, or those with different inductive biases) implement compositional functions. Our findings reflect the tasks we happen to test (often, factual recall) under our specific experimental design. Further work should test other kinds of compositional functions, and try to more deeply understand the relationship between compositional mechanisms, behavior, and generalization.

We investigate a limited subset of mechanisms in language models and use current methods to conduct our analyses. These permit us to decode some, but not all, relevant representational structure. Some signals that we do decode may be a result of feature multiplicity or are not guaranteed to be causal. Finally, some of our tasks (e.g. arithmetic) may be solved by algorithms that we do not consider.

## REPRODUCIBILITY STATEMENT

We make our code fully available so that all of our experiments can be replicated as closely as possible and all computational artifacts (datasets, plots, results) can be reconstructed. We do our best to include all experimental details in the main text and appendices of our paper.

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

## A   DATA CREATION

Table 2: List of our tasks, showing $x$, $g(x)$, and $f(g(x))$ for the random example in Table 1. Tasks with neither $g(x)$ nor $f(g(x))$ are omitted. $f(g(x))$ only shown if distinct from $g(f(x))$.

| $f$ | $g$ | $x$ | $g(x)$ | $f(g(x))$ |
|---|---|---|---|---|
| Word → Antonym | English → Spanish | bogus | false | — |
| Word → Antonym | English → German | philosophical | philosophisch | — |
| Word → Antonym | English → French | excessive | excessive | — |
| x + 10 | 2x | 699 | 1398 | 1408 |
| x + 100 | 2x | 922 | 1844 | 1944 |
| x mod 20 | 2x | 891 | 1782 | 2 |
| Word → Numeric | 2x | one hundred and forty-eight | two hundred and ninety-six | — |
| Word[:-1] | Word[::-1] | responsible | elbisnopser | elbisnopse |
| Rotate(RGB, 120°) | RGB → Name | 8a735a | dimgray | — |

All tasks in Table 2 permit the additional computational pathway $x \rightarrow g(x) \rightarrow g(f(x))$. Those which don't list $f(g(x))$ are commutative and so $f(g(x)) = g(f(x))$ and applying $f$ to $g(x)$ results in $g(f(x))$. The remaining tasks are not commutative, but their formal construction permits the hop $g(x) \rightarrow g(f(x))$ anyway. In particular, $g(f(x))$ equals $g(x)+20$ in `plus-10-times-2`, $g(x)+200$ in `plus-100-times-2`, $g(x) \bmod 40$ in `mod-20-times-2`, and $g(x)$[1:] in `word-truncate-reverse`.

### A.1   TASK CONSTRUCTION

**Antonyms & Translations**   We obtain a list of antonyms from Todd et al. (2024) — further derived from Nguyen et al. (2017) — and obtain translations from Opus-MT (Tiedemann & Thottingal, 2020).

**Factual Relations**   We obtain various factual relations from WikiData and IMDb Non-Commercial Datasets (Vrandečić & Krötzsch, 2014; IMDb.com, Inc., 2024; Bast & Buchhold, 2017). We apply a number of heuristics to obtain well-known and unambiguous mappings. For example, we filter entities by their "sitelinks" on WikiData or "votes" on IMDB (heuristics for popularity) to obtain well-known subjects. To avoid ambiguity, we identify subjects (songs, books, movies, people, etc.) with a single corresponding object (authors, attended universities, etc.). We omit parks and landmarks

that exist in their country's capital. Our exact queries for generating each task can be found in our source code.

**Arithmetic**   We use the range of numbers from 0 to 999 as $x$ in our tasks. These numbers typically result in one token. We use the num2words library to obtain a mapping between words and numeric values. We use the list of antonyms as our list of words for the word-truncate-reverse task.

**Colors**   In the rgb-rotate-name task, we randomly sample RGB colors, rotate them $120°$ by their hue, and map the resulting color value to that color's name (using the webcolors library and the common CSS 3 specification).

## B   IMPLEMENTATION DETAILS

**Examples & Prompts**   We prevent sampling of in-context examples that intersect in the variables $\{x, f(x), g(x), g(f(x)), f(g(x))\}$ with the query. And, as mentioned in Sec. 4.1, we exclude examples in Secs. 4 and 5 which overlap in the first token among their variables. So, although $x =$ "excessive" for the antonym-french task is listed in Table 2, this trivially shares the same first token as $g(x) =$ "excessive" and would be omitted from our analyses.

Our prompts are tokenized differently when predicting numbers or words, e.g. "... \n A: 99" results in [ ][99] whereas "... \n A: modern" results in [ modern]. We accordingly include the trailing space in our prompts when predicting numbers and omit it otherwise. We would then test for the single-token prediction of [99] and [ modern] in this example.

**Representational analysis**   In Sec. 4, we analyze the model's computation from $x \rightarrow g(f(x))$. Consider the query for "Heartbreak Hotel" $\rightarrow$ "1935": i.e. "... Q: Heartbreak Hotel \n A: ". Here, multiple tokens ([ Heart][break][ Hotel][ \][n][ A:][ ]) are central to the computation. We therefore analyze all residual streams for these tokens. At each layer, we measure the signal for each variable by its maximum reciprocal rank across the streams. This procedure yields processing signatures, which quantify the presence of our variables at every layer.

We additionally represent each variable by its first token (since our decoding methods can only produce single-token probabilities) and exclude examples where different variables share the same first token and would be hard to differentiate. For example, $f(x) =$ " modern" and $g(f(x)) =$ " moderno" both share the first token [ modern].

## C   COMPOSITIONALITY GAP BY SIZE

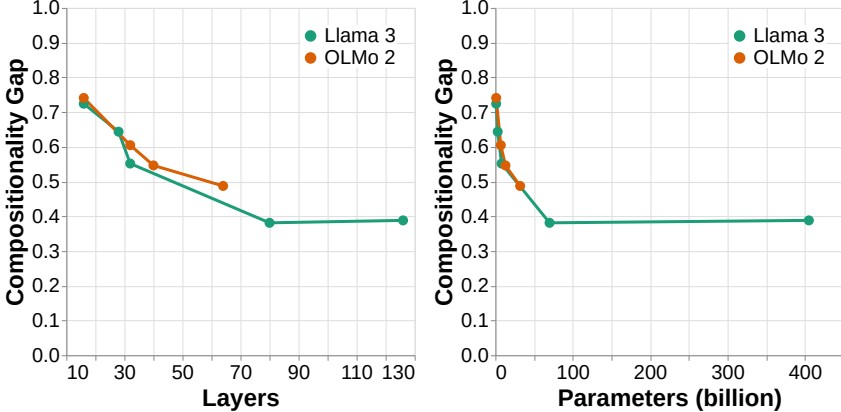

Figure 5: We illustrate the monotonically diminishing improvements to the compositionality gap resulting from increased model size (layers and parameters). We re-visualize results for the OLMo 2 and Llama 3 model families from Fig. 2.

## D    PROCESSING SIGNATURES (CORRECT)

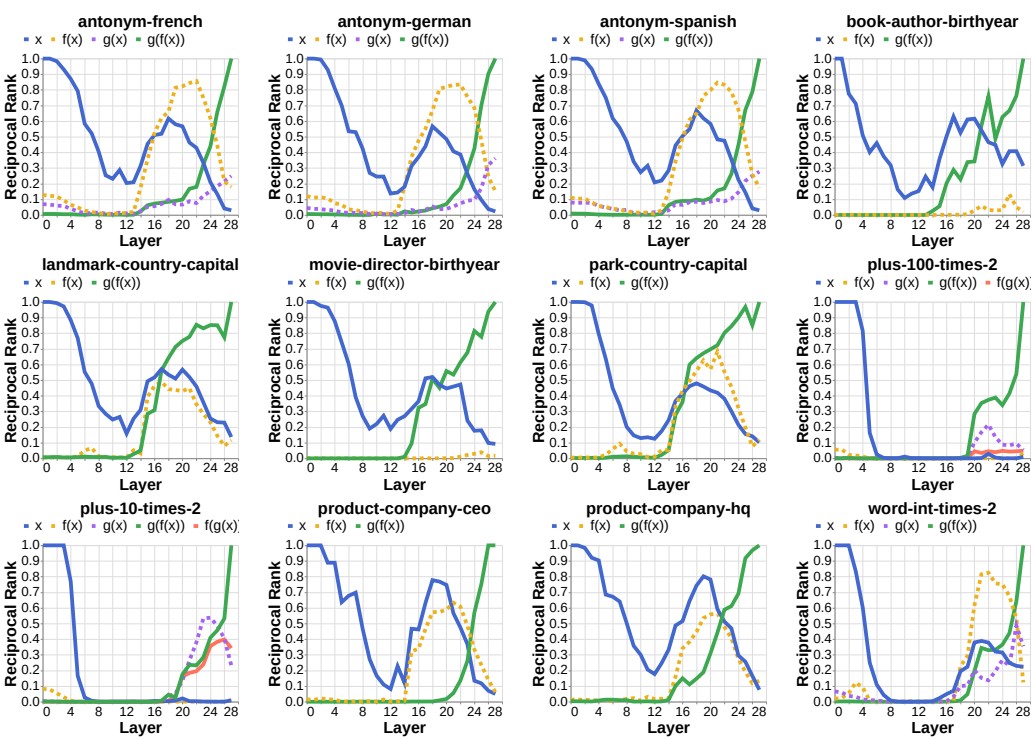

Figure 6: Aggregate processing signatures for each of our tasks, in which Llama 3 (3B) correctly solves all hops and the composition for at least 10 examples.

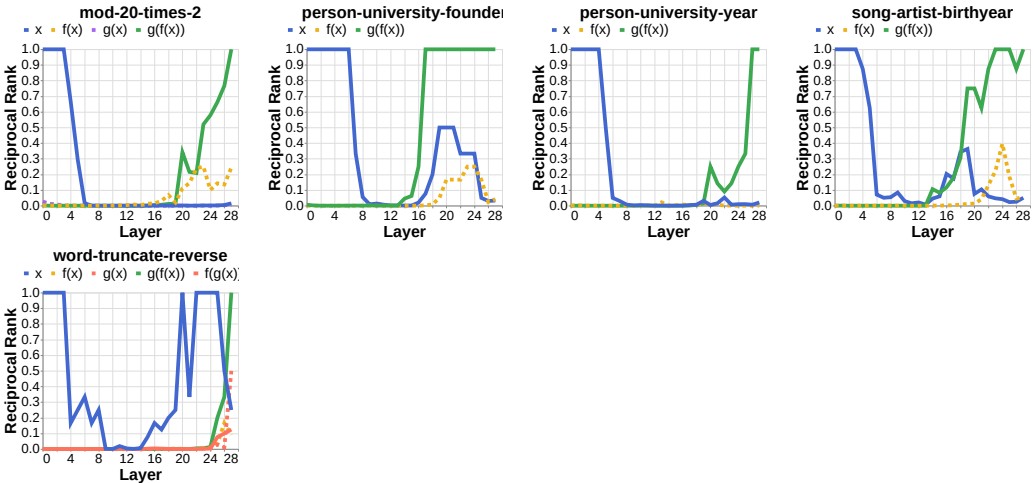

Figure 7: Aggregate processing signatures for each of our tasks, in which Llama 3 (3B) correctly solves all hops and the composition for less than 10 examples.

## E    PROCESSING SIGNATURES (INCORRECT)

Although we see a difference in aggregate processing signatures (Figs. 3a and 3b), where the signal for the intermediate variables is clearer in the correct cases than the incorrect cases, this does not

appear to be generally true (and is more likely due to data imbalances). We can see significant presence of the intermediate variables when considering incorrect examples, de-aggregated by task (Fig. 8).

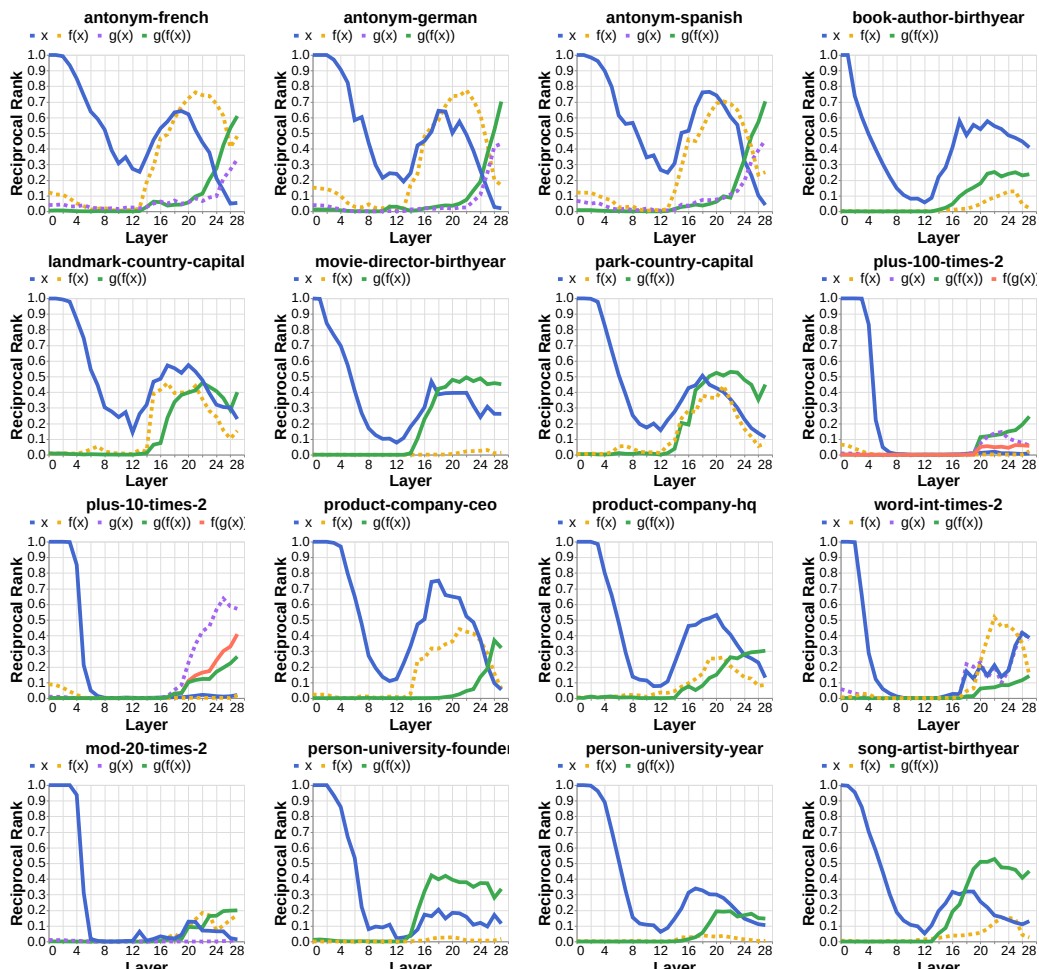

Figure 8: Aggregate processing signatures for each of our tasks, in which Llama 3 (3B) correctly solves all hops but not the composition for at least 10 examples.

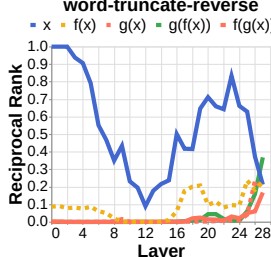

Figure 9: Aggregate processing signatures for each of our tasks, in which Llama 3 (3B) correctly solves all hops but not the composition for less than 10 examples.

# F  TOKEN IDENTITY PATCHSCOPE

Here, we repeat the analyses in Secs. 4 and 5, but use the token identity patchscope (Ghandeharioun et al., 2024) instead of logit lens. This method is proposed as one that is more closely aligned with a language model's computation than other methods (such as logit lens).

We would specifically like to use this method to decode a representation into vocabulary-space logits. To do so, we prompt a model with the "token identity prompt", in which random tokens are repeated twice each, such as "[A] [A] ; [B] [B] ; ... ; [?]". We patch our representation of interest into the residual stream of this forward pass (at the corresponding layer and final token position). The language modeling logits resulting from our intervention then serve as the decoding for our representation.

We generally find similarities with our logit lens analyses: in tasks with "compositional" processing signatures, we continue to see growth of the signals for the intermediate variables with or before that for $g(f(x))$. Please zoom in to observe simultaneous growth, which may be difficult to see due to overlapping lines. And, although these plots may show growth of $f(x)$ and $g(f(x))$ in the same layers, recall that these computations can occur in different (e.g. earlier or later) residual streams (Appendix B).

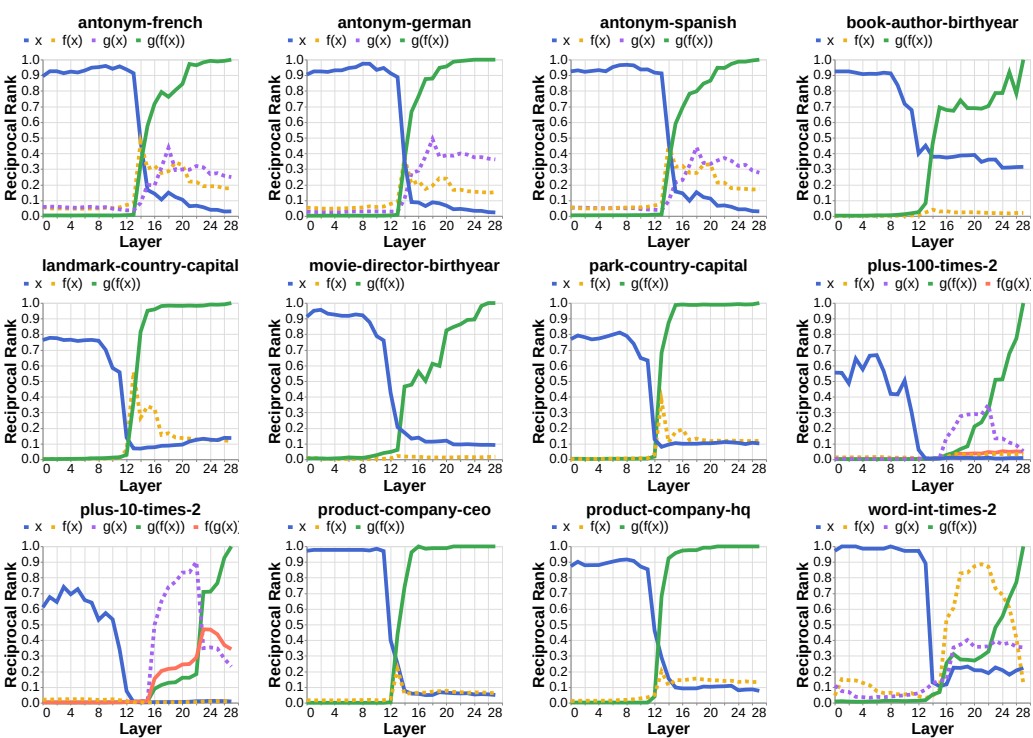

Figure 10: Aggregate processing signatures (using the token identity patchscope) for each of our tasks, in which Llama 3 (3B) correctly solves all hops and the composition for at least 10 examples.

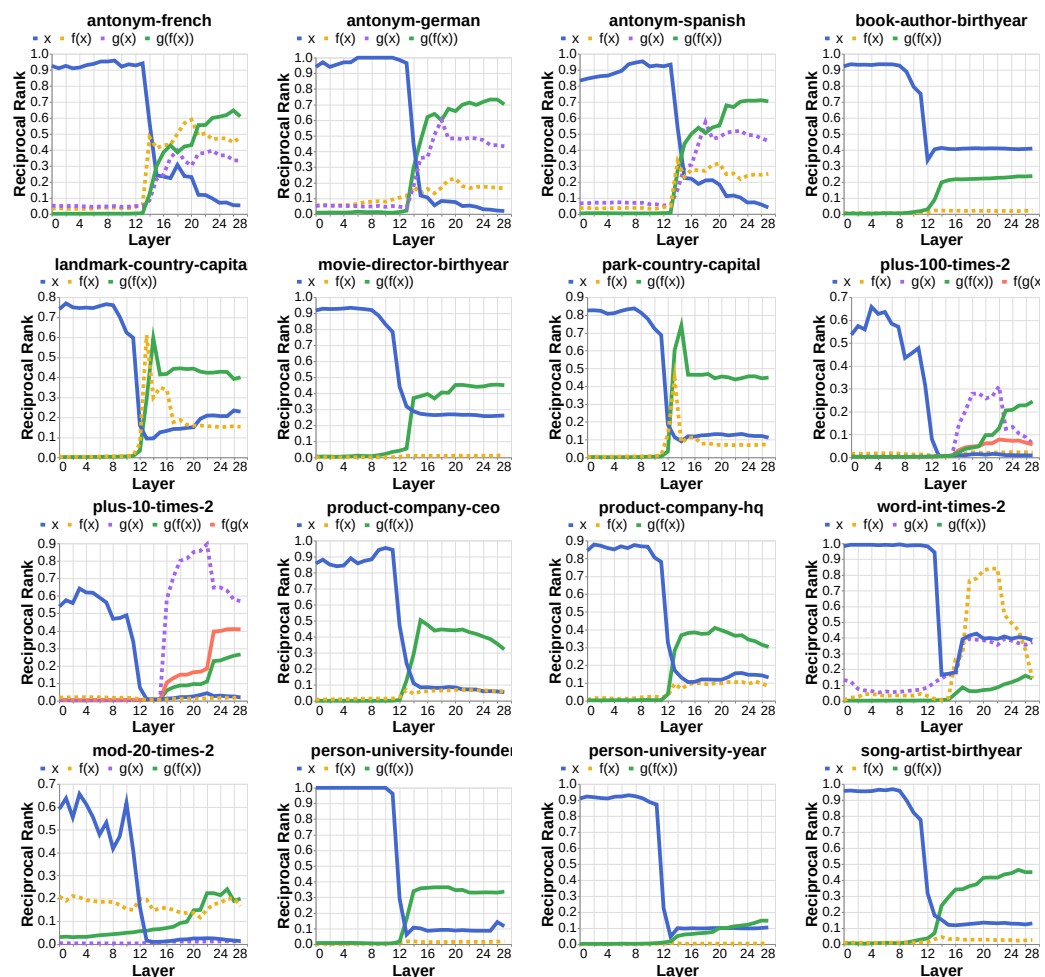

Figure 11: Aggregate processing signatures (using the token identity patchscope) for each of our tasks, in which Llama 3 (3B) correctly solves all hops but not the composition for at least 10 examples.

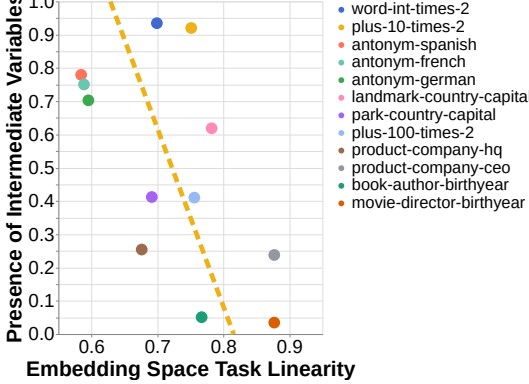

Figure 12: Correlation across tasks ($r^2 = 0.35$) for embedding space task linearity and presence of intermediate variables. Analogous to Fig. 4a, using the intermediate variable metric from the token identity patchscope.

# G    CAUSALITY OF INTERMEDIATE VARIABLES

We would like to determine whether the variables, $f(x)$ and $g(x)$, we identify in models' intermediate representations have a causal effect on the outcome. We describe a preliminary investigation below.

We use activation patching (Vig et al., 2020), a common method for conducting causal interventions in interpretability, and patch representations across tasks.

We first identify tasks with the same $f$ but different $g$, such as antonym-spanish ($g \circ f$) and antonym-german ($g' \circ f$). For some $x$ and $x'$, we extract a single intermediate representation from the forward pass of $g'(f(x'))$ and patch it into the forward pass of $g(f(x))$. On average (over many $x$ and $x'$), we measure the causal effects on the predictions $g(f(x))$, $g(f(x'))$, $g'(f(x))$, and $g'(f(x'))$.

We extract the representation from $g'(f(x'))$ at the position and layer where $f(x)$ or $g(x)$ have the highest reciprocal rank (and only use instances where this value is at least $0.5$). We patch this representation into the forward pass for $g(f(x))$ at the median location where intermediate values are highest (layer 18 and 71st percentile query token position; identified among variables that reach RR $\geq 0.5$). We apply this intervention to two groups: instances with intermediate values that reach a peak RR $\leq 0.2$ and $\geq 0.5$. In other words, instances with direct or compositional processing signatures.

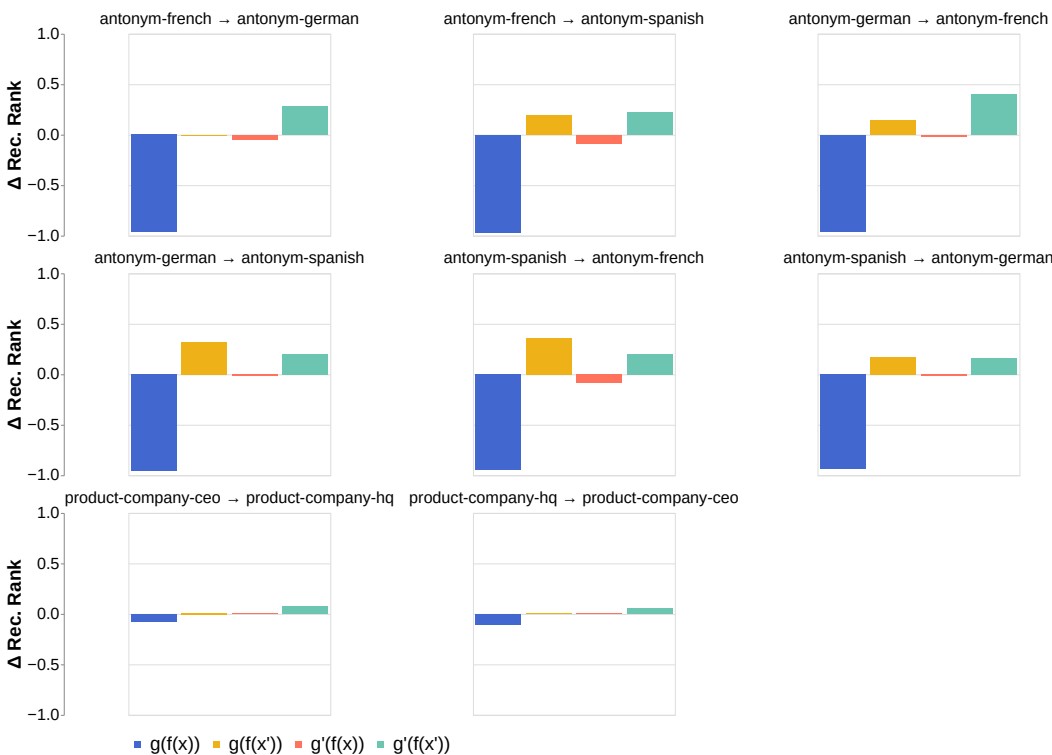

Figure 13: Causal effects on predicted values after patching from $g'(f(x'))$ to $g(f(x))$ for instances with compositional processing signatures.

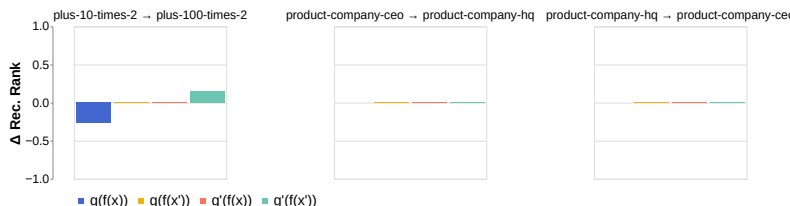

Figure 14: Causal effects on predicted values after patching from $g'(f(x'))$ to $g(f(x))$ for instances with direct processing signatures.

The Antonym–Translation tasks (which tend to have compositional signatures) show the most significant causal effect: on average, $g(f(x))$ and $g'(f(x))$ decrease by -0.95 and -0.4, and $g(f(x'))$ and $g'(f(x'))$ increase by 0.20 and 0.24. The effect on $g(f(x'))$ clearly implicates the existence and causality of $f(x')$ in the patched activation; that on $g'(f(x'))$ indicates the additional existence of either itself or the function vector (Todd et al., 2024) for $g'$ in that representation. The causal effects on compositional instances of `product-company-hq` and `product-company-ceo` are smaller.

But we can also see a clear difference between the causal effects on the compositional and direct instances. Indeed, the effects on `product-company-hq` and `product-company-ceo` are larger in their compositional instances. Patching activations from `plus-10-times-2` into `plus-100-times-2` primarily decreases $g(f(x))$ and increases $g'(f(x'))$, perhaps only implying the existence of the representation for $g'(f(x'))$ in the patched activation.

## H  LINEARITY CORRELATIONS

Similarly to the experiment in Sec. 5 and Fig. 4a, we investigate the relationship between our compositionality heuristic and embedding space linearity for variations of the hops (rather than of the compositional task).

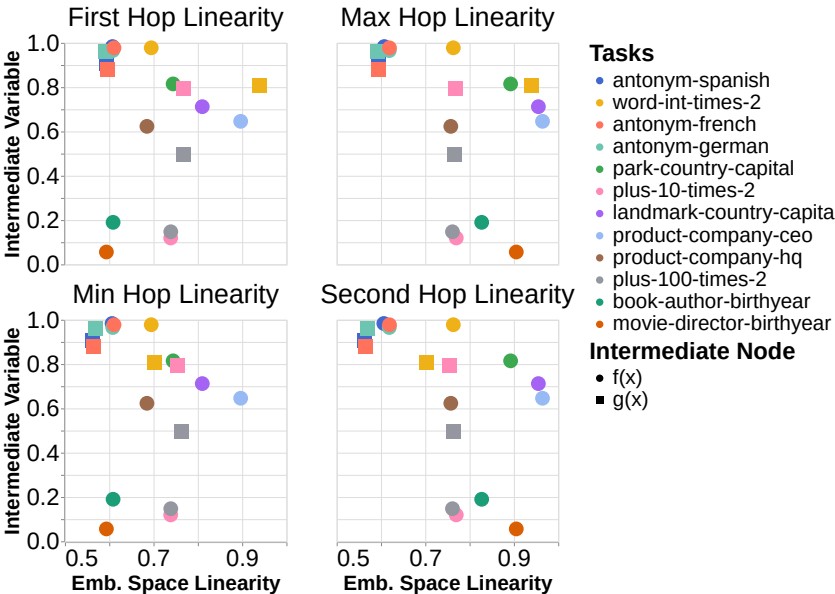

Figure 15: Relationships between presence of intermediate variables and embedding space linearity for the hops. We find weaker correlations in all cases. $r^2 = 0.01$ against the linearity of the first hop; $r^2 = 0.28$ against the second hop; $r^2 = 0.05$ using the minimum linearity between the hops; and $r^2 = 0.20$ using the maximum.

