# OpenReview forum: "How Do Language Models Compose Functions?"
_ICLR.cc/2026/Conference — Submitted to ICLR 2026_

### Official Review · Reviewer_LbGz · 2025-10-28

**Soundness:** 1
**Presentation:** 3
**Contribution:** 3
**Rating:** 4
**Confidence:** 4

**Summary:**

This paper studies the ability of LLMs to apply compositional rules. Recapitulating prior works, they find a "compositional gap": LLMs are often unable to compute composed functions despite being able to compute all of the individual functions in isolation. They find that the compositional gap is shrinking with the use of reasoning models and, contrary to prior works, with the use of larger models, but the gap remains nonetheless. Additionally, they argue that LLMs exhibit two different modes of computing composed functions: *direct* and *compositional*, that is, applying the individual functions together or one after the other. Finally, they argue that the choice of computation mode is modulated by the geometry of the embedding vectors.

**Strengths:**

- It is valuable to see a similar analysis to Press et al. (2022) updated to recent models and reasoning models. To the best of my knowledge, the evidence for a shrinking compositionality gap is novel and very interesting if true. It is also interesting to see that reasoning models still struggle with compositionality, although to a lesser extent.
- The rules appear to be carefully chosen to avoid overlap.
- The distinction between direct and compositional mechanisms is interesting and appears well supported by the empirical analysis

**Weaknesses:**

- The tasks require the LLM to figure out the rule based on 10 given input-output examples. This means that LLMs face the additional obstacle of figuring out the rule. To me, it seems that this additional difficulty is greatly altering the nature of the task and undermining some of the key claims. Intuitively, it seems more difficult to figure out that 2044 is (922+100)*2 than to answer "What is 922 plus 100, times 2?". Under the current prompting formulation, I believe the paper tells us more about ability of LLMs to *learn composed functions in-context* than their ability to *compute composed functions*.
- When the input $x$ consists of multiple tokens, the linearity analysis is performed with the average of all token embeddings. This is an important limitation because, for example, a book title might be composed entirely of words (tokens) that also appear in the titles of another author. We should expect such an average to be equally correlated with the birth years of either authors. However, it is not inconceivable that the LLMs combines the tokens to create an embedding specific to that book and correlated only with the correct birth year. Another approach could be to study the linearity using the residuals obtain when passing $x$ through the LLM (without the in-context examples).
- The literature review is missing recent works on LLM compositionality. For example, works [1-3] also study the ability of LLMs to compose factual and in-context information.

[1] Ni, Ruikang, et al. "Benchmarking and understanding compositional relational reasoning of llms." Proceedings of the AAAI Conference on Artificial Intelligence. Vol. 39. No. 18. 2025.

[2] Musat, Tiberiu. "Mechanism and emergence of stacked attention heads in multi-layer transformers." arXiv preprint arXiv:2411.12118 (2024).

[3] Xu, Zhuoyan, Zhenmei Shi, and Yingyu Liang. "Do large language models have compositional ability? an investigation into limitations and scalability." arXiv preprint arXiv:2407.15720 (2024).

**Questions:**

- Have the authors tried to add the rule to the prompt in natural language, for example "What do you get by adding 100 and then doubling?", "What is the birth year of the author of this song?", etc., in addition to the 10 examples ? Does the narrowing of the gap with increasing scale persist in this regime?
- Have the authors attempted to compute the linearity using the residual stream of the input $x$?
- Do the authors have any insights on how can it be that reasoning models still exhibit a compositional gap? It seems that they should be able to just apply the hops one by one using the reasoning tokens.

---

> ### Author Response · Authors · 2025-11-23
>
> Thank you for your review! We appreciate your ideas and would be happy to elaborate further.
>
> - **[W1 & Q1]** You raise an interesting point and we’d be happy to discuss it further. As we see it, our research question is if/how models break compositional tasks into sub-tasks: do models solve $x \to y$ via $g(f(x))$ or $(g \circ f)(x)$? Although humans might prefer the former, there is no reason a neural network must represent a task either way (i.e. as primitive functions $\{f, g\}$ vs. a compositional one $g \circ f$). It’s actually a common criticism that neural networks might approximate functions without actually encoding the internal structure. So we do not explicitly prompt the model with information about the task's structure: this is a control to avoid biasing the model’s representations towards the compositional structure [Footnote 4]. The study you propose is also interesting, but it answers a different question. We also agree that as part of our task, models must infer the rule from $x \to y$ examples. This is by design, since ICL is a natural way to seed a model with a task without explicitly prompting it with the task’s structure. Even so, our analyses in Sec. 4 and 5 focus on the latent computation corresponding to $x \to y$, rather than the computation that concerns inferring the task from examples. For this reason, we believe our work is indeed about *how* models “compute composed functions”.
>
> - **[W2 & Q2]** This is another great question. We’d first like to refer you to work that does find linear relationships between residual stream activations and unembeddings [1]. Our work is intentionally narrower: we study linear structure purely at the level of model *primitives* (i.e. embeddings and unembeddings). We find it compelling to relate models’ internal mechanisms directly to the geometry of these fundamental building blocks. With this goal, our pooled representation of input embeddings is indeed simple and order-agnostic, and we agree it is not a perfect representation of the target concept. We would be happy to hear suggestions for other ways to form this representation from primitives alone. Nonetheless, we still find strong correlations in the downstream phenomena, which shows these representations are at least sufficient for our purposes. We also agree that having different inputs with permuted tokens is problematic in theory, but don’t find this representative of our data in practice.
>
> - **[W3]** Thanks for bringing those citations to our attention. We’d be happy to mention them in our paper.
>
> - **[Q3]** We completely agree that reasoning models have the means to solve these tasks *in theory* (as you say, by computing hops in each generated token). In practice, existing reasoning models seem to deviate from this ideal behavior. While this is an important question, we feel that it would be better addressed by future work, per [L463-464], for a few reasons. (1) The reasoning traces we have from DeepSeek-R1 are up to 2000 tokens long. It’s a rather large task to analyze these and make general claims. (2) As of the latest research, it is still unclear whether the traces from chain-of-thought or reasoning models are actually faithful to models’ predictions [2, 3]. (3) Current interpretability methods are not yet suitable for understanding the (sizable) latent computation in reasoning models.
>
> [1] Evan Hernandez, Arnab Sen Sharma, Tal Haklay, Kevin Meng, Martin Wattenberg, Jacob Andreas, Yonatan Belinkov, David Bau. Linearity of Relation Decoding in Transformer Language Models. ICLR 2024.
>
> [2] Tamera Lanham, ..., Jared Kaplan, Jan Brauner, Samuel R. Bowman, Ethan Perez. Measuring Faithfulness in Chain-of-Thought Reasoning. 2023.
>
> [3] Anthropic. Reasoning Models Don’t Always Say What They Think. 2025.

---

### Official Review · Reviewer_NP7L · 2025-10-29

**Soundness:** 2
**Presentation:** 3
**Contribution:** 1
**Rating:** 2
**Confidence:** 3

**Summary:**

The authors inspect LLM's ability to process composed functions x->f(x) and g(f(x)) versus directly computing x->g(f(x)) in a 'one-shot' fashion. Upon inspecting the LLM's inner mechanisms they find that LLMs contain both, individual circuits for f and g and direct shortcut circuits $g\circ f$. The authors conclude that direct shortcut mechanisms are leveraged whenever there exists a linear mapping between the input and output representation.

The authors present an empirical study on compositional reasoning, evaluating how different LLMs perform on tasks requiring said capacity. One of the main points is that LLMs often make use of idiomatic reasoning instead of compositional reasoning. In detail, a number of experiments is performed to support two statements: 1) Within the limits of current methodologies for interpreting LLM's inner workings, compositionality remains an open challenge. The performed experiments supports that current LLMs do not show hints of compositional reasoning for many tasks that clearly require it. 2) The geometry of the embedding space plays an important role. Experiments hint that the presence of compositionality is inversely correlated to the linearity of the embedding space.

**Strengths:**

The presented research question is of high importance for understanding the inner workings of LLM reasoning. Similar to previous works, the authors demonstrate a gap between compositional and direct short-cut reasoning of LLM. The gap is shown across models of different sizes, including instruction tuned and thinking models.

The the paper is well structured and easy to read. Results are clearly presented and discussions align with the obtained results. To the best of my knowledge the authors soundly apply existing techniques and reasonably cover and discuss related work.

Findings on embedding space linearity pose new insights on LLM reasoning, allowing the prediction of the inspected phenomena beyond the inspected models.

**Weaknesses:**

1) The authors seem to exclusively employ rather outdated "Q:\<query\>\\nA:" in-context learning prompts for pretrained autoregressive language models in their experiments. For the tested thinking and instruction tuned models, the paper is lacking details on the exact prompt structure. Additionally, I could not find any examples of the exact prompts used to embedding the tasks which could be a major factor in model performance.
2) More generally, the authors give no insights into the observed failure modes of the models, e.g., none of the obtained outputs are shown or analyzed. The authors mention that results are only considered under exact answer matching which might reduce accuracy, due models not adhering to the expected format. Similarly, reasoning models are mentioned to be given a 2000 characters limit. While this might be a sufficient for testing the rather simple tasks, it could be that decreased accuracy in the compositional case might simply stem from aborting due to the token limit.
3) The employed logitlens and patchscope techniques, which establish main findings of the paper, are not explained. No discussion on the assumptions or limitations of these techniques is made, which requires the reader to already have a deep understanding of these methods. As mentioned above, details on prompts and result evaluation are generally missing, making the paper not self-contained and lack in technical detail.
4) While the authors initially demonstrate a compositionality gap on multiple recent and instruction-tuned models, the main evaluation seem to only be performed on the rather old and weakly performing Llama 3-3B. It is unclear to me, whether the conclusions gained from these evaluations would generalize to modern LLM. Given the additional lack in prompt descriptions and introspection techniques, I find it hard to judge the significance of the presented work.

**Questions:**

I would like to ask the authors to elaborate on the following points:

1) How do the exact prompts templates and obtained results look like? Have the authors made sure that performance degradations do not stem from insufficient prompt embedding of the tasks or mismatches in the output format? Where there any commonly observed failure modes that could have biased results?
2) How do the obtained insights transfer to modern instruction-tuned or 'thinking' LLMs?
3) How exactly did the authors modify the evaluation mentioned in sec. 3.1? How do the authors determine a lexical unit in this particular setting?
4) What the the assumptions made by logitlens and patchscope and do these assumptions hold in the presented setting? Are the mentioned logitlens and patchscope techniques applicable to longer reasoning traces?

---

> ### Author Response · Authors · 2025-11-23
>
> Thank you for your review! We appreciate that you agree about the importance of our research question and insights. We’d like to answer your questions below.
>
> - **[W1 & Q1]** Let’s consider our research question again: when a model computes a (compositional) function, does it do so via $(g \circ f)(x)$ or $g(f(x))$? Our experimental design is entirely motivated by our question. ICL enables us to seed the model with this function from $x$ and $y$ alone. We don’t additionally “embed tasks” in the prompt to avoid explicitly biasing the model towards either $g \circ f$ or $\{g, f\}$ (Footnote 4).
> We then intentionally use a simple and standard ICL prompt (the same as in [1]). We consider this prompt “long-standing” rather than “outdated” as we expect modern models to be robust to such standard ICL settings. It is compelling that we demonstrate consistent, negative results on the compositionality gap under these well-known settings. We believe our chosen prompt is a suitable representative: we also ablate our Llama 3 (3B) model (same settings as Fig. 2) with [many alternative ICL prompts (from [1] App. C, Table 8) and find rather similar performance in all cases](https://anonymous.4open.science/api/repo/openreview-h7ZAgetMLc-4EB2/file/compositionality_gap_by_prompt.pdf?v=4f60db9b). Finally, the prompts are the same for all models, including for instruction-tuned and reasoning models. There is no further information about the prompts omitted from our paper.
>
> - **[W2 & Q1]**
>   - **Analyzing Failures:** We’d be happy to include a random sample of correct/incorrect model predictions in our appendix and, for now, [do so here](https://anonymous.4open.science/api/repo/openreview-h7ZAgetMLc-4EB2/file/predictions_all_models.html?v=198ce0ac) (showing up to 5 samples per model/task). We do actually show processing signatures of failures in App. E. These signatures reveal several deeper insights about *why* models fail (e.g. compared to more qualitative, surface-level analysis of model outputs). We can see many tasks in which the model computes the first hop, but fails to apply the second function. For example, “antonym-french” shows $f(x)$ diminishes more slowly after peaking than in correct cases (App. D) and subsequently applies $g$ more slowly. “plus-10-times-2” shows $g(x)$ is computed “late” (peaking at layer 25 vs 23) and dominates the prediction, and that $f(g(x))$ is preferred to $g(f(x))$, in which case $f$ is incorrectly applied after $g(x)$. We will be sure to further highlight these insights in our paper.
>   - **Evaluation:** We consider producing the most likely continuation (i.e. the simple, exact span that should follow “A:” in the ICL prompt) as part of our task. (We even use the Completions API, rather than Chat Completions, to specifically measure this in API-based models.) Since we specifically measure the compositionality gap, we also have an expectation that the model is already able to solve the underlying hops under the same exact match metric. For these reasons, we believe it is acceptable to use exact match accuracy as part of our evaluation. If we were measuring the overall performance on solving compositional tasks, this would be less appropriate. Lack of adherence to the expected format is not a common failure mode: this only occurs to a significant degree in o4-mini on “plus-100-times-2”. If we use a less conservative evaluation and accept these answers, this would result in a 6% (abs) reduction in compositionality gap for o4-mini alone. Our paper’s claims are unaffected either way, but we will mention this consideration in the appendix.
>   - **Reasoning budget:** We would like to clarify that the reasoning budget is 2000 tokens [L194-195], not characters; this is approx. 1,500 words. We believe it is a very liberal budget given that we simply test 2-hop factual retrieval tasks; it is also the default setting on OpenRouter. Finally, this reasoning chain and budget are independent of the continuation. So even if the model does not finish reasoning, it would still produce a continuation rather than aborting.
>
> - **[W3 & Q4]** We explain logit lens in [L221-224], the token identity patchscope in [L866-874], and their limitations in [L468-470]. Restating their core limitations: these methods decode some, but not all, relevant representational structure and it is possible for their decoded representations to be spurious features. The former is why we test with both methods and the latter is why we demonstrate causal significance of intermediate variables in App. G. It would not be easy to analyze long reasoning chains using these methods, simply because it is difficult to analyze the 10,000s of representations across 100s of token positions. New interpretability tools are necessary for this difficult task of understanding the latent computation in reasoning models. Even though reasoning augments the capabilities of models, our work does study the principle computation that underlies reasoning.

---

> > ### Author Response · Authors · 2025-11-23
> >
> > - **[W4 & Q2]** We agree that extending our analyses to other models would further demonstrate the generality of our findings. We have repeated all of our main body experiments on OLMo 2 (7B & 13B), as well as the instruction-tuned variant of Llama 3 (3B). Our new results equally or even more strongly support our findings and hypothesis! We continue to observe bi-modality of processing mechanisms (Sec. 4) and evidence for the linearity–mechanism relationship (Sec. 5). OLMo 2 (7B) correlates with [$r^2 = 0.75$](https://anonymous.4open.science/api/repo/openreview-h7ZAgetMLc-4EB2/file/correlation_olmo_2_7b.pdf?v=4cd2c3d9), OLMo 2 (13B) correlates with [$r^2 = 0.50$](https://anonymous.4open.science/api/repo/openreview-h7ZAgetMLc-4EB2/file/correlation_olmo_2_13b.pdf?v=adb29ea1), and Llama 3 (3B) Instruct correlates with [$r^2 = 0.63$](https://anonymous.4open.science/api/repo/openreview-h7ZAgetMLc-4EB2/file/correlation_llama_3_3b_instruct.pdf?v=f46e245c). We note that larger models and reasoning models are currently out-of-scope for interpretability work, due to computational and methodological limitations, and that Llama 3 (3B) was released in September 2024 (so we don’t consider this model particularly old).
> >
> > - **[Q3]** To clarify, we conduct our evaluation using the settings our paper describes. We then find a compositionality gap, which is a phenomena first observed by [2]. That is to say: we do not start with the methodology in [2] and modify it; our work considers separate settings from [2]. [“Lexical units”](https://en.wikipedia.org/wiki/Lexical_item) are defined as a few words that convey a single meaning. The values of our datasets’ variables ($x$, $f(x)$, …) are lexical units: names, birth years, numbers, words, and so forth.
> >
> > [1] Eric Todd, Millicent L. Li, Arnab Sen Sharma, Aaron Mueller, Byron C. Wallace, David Bau. Function Vectors in Large Language Models. ICLR 2024.
> >
> > [2] Ofir Press, Muru Zhang, Sewon Min, Ludwig Schmidt, Noah A. Smith, Mike Lewis. Measuring and Narrowing the Compositionality Gap in Language Models. EMNLP 2023.

---

> > ### Comment · Reviewer_NP7L · 2025-11-26
> >
> > Dear authors,
> > thank you for the detailed and thoughtful rebuttal.
> >
> > I'm happy to see the experiments on different models and prompts, as also requested by the other reviewers, to hold similar results and confirm the findings of the initial experiments. The provision of example prompts and replies relieves concerns on out-of-token failures or getting stuck in repetition loops, strengthening my belief of a sound evaluation.
> >
> > Also, thanks again for pointing out the discussion on logit lens and token identity patchscope. I was expecting to find a more formal treatise of the methods and might have overlooked the existing discussion. I believe the current state might be sufficient.
> >
> > Even after your initial rebuttal I find a critical open point remaining, that concerns the few-shot CoT prompting via the conditioning on example question and answer in the style of [1]. On that, I concur with Reviewer LbGz on that with the current setup it is unclear, whether the authors are actually testing the compositional reasoning of LLM or whether performance losses are due to the ability of the models to infer of the correct rule from the provided examples. Even though the previous inspection method might not be directly be applicable, providing such an evaluation with rules given in natural language would give an indication of whether the observed performance losses are due to the claimed reasoning gap or due to the failure of rule inference.
> >
> > Since all other point have been cleared I have raised my score to a marginally reject. However, I am still skeptical due to the remaining concern, but happy to discuss this possible point further.
> >
> >
> > [1] Wei, Jason, et al. "Chain-of-thought prompting elicits reasoning in large language models." Advances in neural information processing systems 35 (2022): 24824-24837.

---

> > > ### Author Response · Authors · 2025-12-02
> > >
> > > Thanks to the reviewer for their continued engagement! We are glad we could resolve nearly all concerns. On this final point:
> > >
> > > > concerns the few-shot CoT prompting [experiments] ... in the style of [1] ... concur with Reviewer LbGz
> > >
> > > We'd like to clarify that our experiments don't involve CoT prompting — LbGz was just referring to the idea of embedding a task instruction in the prompt.
> > >
> > > > concur with Reviewer LbGz [that] it is unclear whether the authors are actually testing the compositional reasoning of LLM or whether performance losses are due to the ability of the models to infer the correct rule from the provided examples
> > >
> > > In our task, the model must infer a function $F$ from the prompt and apply $F$ to the query — whether $F$ is $f$, $g$, or $g \circ f$. The claims our paper makes are not about the overall performance for solving compositional tasks $(g \circ f)(x)$. Instead, we specifically measure the compositionality gap: can the model solve $(g \circ f)(x)$, given that it can solve the hops? We know the model is already able to infer $f$ from examples $x_i \to f(x_i)$ and $g$ from $f(x_i) \to (g \circ f)(x_i)$. Failure to infer $g \circ f$ from $x_i \to (g \circ f)(x_i)$ is consequently a failure in compositional ability for our task. In other words, we consider failures in either function inference or application as contributors to the compositionality gap.
> > >
> > > In Sec. 3, our work demonstrates that a certain class of tasks (i.e. two-hop tasks under standard ICL settings) consistently exhibit a compositionality gap across modern models — we investigate this phenomena further in the following sections, in alignment with our research questions. The proposed idea is very interesting, but addresses a different research question: how does prompting models with task instructions affect their compositional abilities? We'll emphasize that, even in this case, the model must still infer $F$ from the prompt (whether via examples or instructions).
> > >
> > > Since there is reviewer interest, we've conducted additional experiments to investigate this question. We repeat our experiments for Fig. 2 *, but always prefix the existing ICL prompt with a task instruction: e.g. "Add 100.\n\n" for $x \to f(x)$ or "Add 100, multiply by 2.\n\n" for $x \to g(f(x))$. We continue to find a [clear compositionality gap in our results](https://anonymous.4open.science/api/repo/openreview-h7ZAgetMLc-4EB2/file/compositionality_gap_instruction_prompts.pdf?v=7ab23df1). We observe the gap diminishes more steeply w.r.t. model size or reasoning than in the original experiments. Llama 3 (70B) is also substantially closer to closing the gap. In any case, our original claims pertain to the common setting we do investigate (rather than all classes of compositional tasks) and are independent of these particular findings.
> > >
> > > \* Note: we omit the instruction-tuned Llama model and OpenAI models, whose outputs are harder to constrain (and thus evaluate) given the additional instruction prompt.

---

### Official Review · Reviewer_zRpK · 2025-10-29

**Soundness:** 3
**Presentation:** 3
**Contribution:** 3
**Rating:** 6
**Confidence:** 4

**Summary:**

This paper investigates how large language models (LLMs) perform compositional reasoning , whether they explicitly compute intermediate steps when solving tasks of the form g(f(x)). The paper introduces a set of new two-hop compositional factual retrieval tasks that can be expressed in the form of g(f(x)). These tasks focus across various domains, including arithmetic, translation and rotation. Based on these tasks, the authors confirm that current LLMs still suffer from a “compositionality gap”, which they define as the ability to correctly answer each individual hop (computing z=f(x) and y=g(z)) but not their overall composition of g(f(x)). Using a logit lens analysis of residual stream activations, the authors find two distinct processing mechanisms. In some tasks, models explicitly represent intermediate variables (a compositional mechanism), in others, they rely on a direct or “idiomatic” shortcut, enabled when a near-linear mapping exists between x and g(f(x)) in the embedding space.

**Strengths:**

1. The paper is well motivated: it systematically explores whether large language models perform compositional reasoning explicitly or implicitly through function composition tasks
2. The paper is well-structured: it first quantifies the compositionality gap using a newly introduced dataset of two-hop factual tasks, then analyzes the internal mechanisms through a logit lens, and finally attempts to explain the observed behavior.
3. The finding of the paper is interesting: each section reveals distinct and insightful aspects of factual knowledge encoded in LLMs.

**Weaknesses:**

1. The evaluation is limited in scope: as the paper focuses on a single, smaller model from the Llama 3 family (3B). Including larger models and models from different families would help validate and strengthen the paper’s conclusions. Performing experiments on opensource models with opensourced lenses like Pythia, Gpt2XL, Llama2 would help make the results more generalizable. The authors could potentially present these during the rebuttal.
2. Prompt design for section 3: the paper only used one simple in-context prompting method. It would be interesting to see by varying prompting strategy (such as chain-of-though) how would it affect the performance of the model
3. The interpretation in Section 5 is limited: as the paper only reports the correlation (r²) between embedding linearity and compositional processing. However, correlation does not necessarily imply causation. Hence, additional causal evidence or directional analysis would strengthen the argument. Additionally, testing out at least one more reasonable hypothesis that gets negated would strengthen the paper.

**Questions:**

Look at weakness section

Figure 1 is slightly hard to understand. Might be better to separate the two.

Instead of writing All hops, would be nice to specify 1 hop, 2 hop, 1+2 hops.

---

> ### Author Response · Authors · 2025-11-23
>
> Thanks for your review and feedback! We’d be happy to discuss further below.
>
> - **[W1]** We completely agree that extending our analyses to another model family would further demonstrate the generality of our findings. We have repeated all of our main body experiments on OLMo 2 (7B & 13B), as well as the instruction-tuned variant of Llama 3 (3B). Our new results equally or even more strongly support our findings and hypothesis! We continue to observe bi-modality of processing mechanisms (Sec. 4) and evidence for the linearity–mechanism relationship (Sec. 5). OLMo 2 (7B) correlates with [$r^2 = 0.75$](https://anonymous.4open.science/api/repo/openreview-h7ZAgetMLc-4EB2/file/correlation_olmo_2_7b.pdf?v=4cd2c3d9), OLMo 2 (13B) correlates with [$r^2 = 0.50$](https://anonymous.4open.science/api/repo/openreview-h7ZAgetMLc-4EB2/file/correlation_olmo_2_13b.pdf?v=adb29ea1), and Llama 3 (3B) Instruct correlates with [$r^2 = 0.63$](https://anonymous.4open.science/api/repo/openreview-h7ZAgetMLc-4EB2/file/correlation_llama_3_3b_instruct.pdf?v=f46e245c).
>
> - **[W2]** That’s a good point and we did consider it. We believe that chain-of-thought is one way (by prompting) to leverage additional autoregressive generation (i.e. recurrent computation) in a model. Reasoning models do the same but leverage more recent methods and offer higher performance. So we opted to test the latter. We more generally try to be careful with our prompting (e.g. as a control for our analyses, we don’t explicitly prompt models with information about $f$ and $g$ [Footnote 4]). But within these constraints, we also ablate our Llama 3 (3B) model (same settings as Fig. 2) with [many alternative ICL prompts (from [1] App. C, Table 8) and find rather similar performance in all cases](https://anonymous.4open.science/api/repo/openreview-h7ZAgetMLc-4EB2/file/compositionality_gap_by_prompt.pdf?v=4f60db9b).
>
> - **[W3]** We agree that finding causal evidence would be very interesting. In fact, we originally hoped to include experiments to confirm the causal effect. However, we found that it was hard in practice to induce linearity in the embedding space without re-pretraining the model. We attempted to fine-tune the model (updating the whole model or just the embedding matrix; 10,000 steps, varying batch sizes and LRs) on our non-linearly-represented tasks, but were unable to induce any significant change in linearity. The model was likely able to fit well with shallower updates. [2] does find a correlation between co-occurrence frequency and representational linearity, but determining causality would likely require manipulating the training data and pre-training models. For now, we present strong correlational evidence to support the relationship. We are working on new experimental designs to test this, we’d love any specific ideas/insights you might have for how to manipulate this in already trained models! Separately, we do present causal evidence for the role of the mechanism in applying consecutive functions in App. G. Finally, we do refute other hypotheses, including: relationships for the mechanism with accuracy [L249-252], for the mechanism with the linearity of the hops [App. H], and for linearity with accuracy [L339-340].
>
> [1] Eric Todd, Millicent L. Li, Arnab Sen Sharma, Aaron Mueller, Byron C. Wallace, David Bau. Function Vectors in Large Language Models. ICLR 2024.
>
> [2] Jack Merullo, Noah A. Smith, Sarah Wiegreffe, Yanai Elazar. On Linear Representations and Pretraining Data Frequency in Language Models. ICLR 2025.

---

### Official Review · Reviewer_KcBP · 2025-11-01

**Soundness:** 2
**Presentation:** 2
**Contribution:** 3
**Rating:** 4
**Confidence:** 3

**Summary:**

This paper investigates the question of how language models handle compositional processing. Specifically, they first confirm that modern LMs (at least Llama-3 3B) continue to suffer from the compositionality gap identified in previous literature, in which for some task expressable as g(f(x)), they can compute f(x) and g(z) separately but not the full problem. They then use logit lens in order to identify a compositional and idiomatic path to solving problems expressable in this form, and find that the idiomatic mechanism may be dominant in cases where there is a linear mapping from x to g(f(x)) in embedding space.

**Strengths:**

**S1**: I appreciate the thorough discussion of related literature, especially in the philosophical and cognitive science domains. I like that the paper thoughtfully engages with this literature and positions findings relative to previous work on compositionality and debates about compositional behaviour vs. representation.

**S2**: The framing and experimental setup are very clear and examination of the basic form of f(g(x)) is very well explored through many different types of relationships. I also think the finding that the idiomatic path is more prevalent when there’s a direct linear mapping is very interesting and the different categories could also shed some light on why (for instance, for factual questions in many cases the answer could be mentioned directly while for computational questions it would be rare to see exact intermediate steps in training data).

**Weaknesses:**

**W1**: In general, it seems like the setup could have been expanded a bit in a way that would make the results more generalizable and robust. Although this paper is an analysis paper and is not meant to probe complex reasoning, I think that a few additional experiments or tweaks would help make the results much stronger:
- The decision to only use a single token prediction seems overly strict, as several tasks with multi-token outputs are excluded from the analysis. For instance, x = “excessive” shares the same token as g(x) = “excessive”, but it seems like this would unnecessarily exclude many possible compositions. Some effort should be made to change the setup to be usable at the span level through pooling results from logit lens if possible.
- Although many models are explored for the compositionality gap results, there is only a single model (Llama3 3B) tested for the main results. It would be interesting and speak to the generality of the results if at least one other model family at a different size was also tested.

**W2**: While the paper addresses an interesting theoretical question about compositionality, the practical implications of the findings remain unclear. The experimental setup largely adapts existing methods (compositionality gap from Press et al. 2022, processing signatures from Merullo et al. 2024), and the novel contribution—the correlation between embedding space linearity and processing mechanism—lacks clear connections to downstream outcomes. How should these findings inform model development, evaluation strategies, or our understanding of when models will succeed or fail at compositional reasoning?

**W3**: This is a minor weakness, but just putting some suggestions here: the figures in this paper could be improved in clarity and visuals,

- It was very hard to interpret Figure 1, the upside down axis for absolute is confusing. To make this clearer I would suggest just having the red bar made up of the relative proportions of the yellow and blue bars but of different heights, or to simply make this two separate figures instead of trying to stay on the same axis. The color labels are also confusing, from what I understand yellow is P(final correct | all hops correct) and blue is 1 - P(final correct | all hops correct) but I’m still not sure I interpreted this correctly.
- Figure 2 should not be a line plot. The x axis values are different models and while they do increase in size, making this a line plot implies some kind of continuous relationship. Additionally the dual axes are again hard to interpret with compositionality gap vs. proportion correct, I think it would also be clearer to split this into two figures.

**Questions:**

- is the linearity-mechanism relationship causal? Can you test whether manipulating linearity causes more idiomatic processing, or whether both arise from a common factor such as frequency of the direct mapping in training data?

- does processing mechanism predict whether an LM generalizes compositionally? It seems like the compositional path is more general but it seems like it could be fine to just memorize some direct mappings especially for QA and this may not hurt performance.

- why limit to single-token outputs? Is there a reasonable way to get around this, given that this excludes many natural tasks?

---

> ### Author Response · Authors · 2025-11-23
>
> Thank you for your thoughtful review and feedback! We’d be happy to address your points below.
>
> - **[W1b]** We completely agree that extending our analyses to another model family would further demonstrate the generality of our findings. We have repeated all of our main body experiments on OLMo 2 (7B & 13B), as well as the instruction-tuned variant of Llama 3 (3B). Our new results equally or even more strongly support our findings and hypothesis! We continue to observe bi-modality of processing mechanisms (Sec. 4) and evidence for the linearity–mechanism relationship (Sec. 5). OLMo 2 (7B) correlates with [$r^2 = 0.75$](https://anonymous.4open.science/api/repo/openreview-h7ZAgetMLc-4EB2/file/correlation_olmo_2_7b.pdf?v=4cd2c3d9), OLMo 2 (13B) correlates with [$r^2 = 0.50$](https://anonymous.4open.science/api/repo/openreview-h7ZAgetMLc-4EB2/file/correlation_olmo_2_13b.pdf?v=adb29ea1), and Llama 3 (3B) Instruct correlates with [$r^2 = 0.63$](https://anonymous.4open.science/api/repo/openreview-h7ZAgetMLc-4EB2/file/correlation_llama_3_3b_instruct.pdf?v=f46e245c).
>
> - **[W1a & Q3]** This might refer to two points.
>   1. We actually do include examples with multi-token outputs in our analyses: we represent these using their first token [L731-734]. Since we expect the computation of interest $x \to g(f(x))$ to occur within this first-token prediction, our existing logit lens setup is suitable for these cases as well.
>   2. We indeed exclude examples where variables share the same first token. This is a critical control to ensure that we correctly differentiate between signals for variables in the latent computation. E.g. in the example you mention, how would we know whether the logit lens value for “excessive” should be attributed to $x$ or $g(x)$?
>
> - **[W2]** While we agree that practical and actionable insights are the ultimate goal, our specific study is an analysis paper and so these are not our immediate goal. That said, we do see practical paths forward toward model building and evaluation. For example, our work identifies which functions language models consider “primitive” from embeddings alone. Models dedicate their representational capacity towards memorizing these primitive functions. If one believes that mechanisms are requisite to generalization (an open question), they may prefer a model that fits to truer primitives ($f$, $g$, and $h$ rather than $g \circ f$ and $h \circ g$) and invokes the compositional mechanism. This would permit computing unseen compositions, say $f \circ h$. Our work opens an avenue for future efforts towards this geometry and mechanism (e.g. via inductive biases for composition, like in optimization or architecture). For evaluating compositional failures, our processing signatures identify which functions the model attempted to invoke ($f$ and $g$ vs $g \circ f$ — as seen for failure cases in App. E) and which hops are successful. We could also see when a particular data point $x$ fits poorly to a function’s linear mapping in the embedding spaces, indicating poor memorization during training.
>
> - **[W3]** Thanks for the input on our figures. Here’s a [new proposal of Fig. 1](https://anonymous.4open.science/api/repo/openreview-h7ZAgetMLc-4EB2/file/compositionality_gap_2_bars.pdf?v=aafaa6d0) with labels following your P notation. Hopefully, that’s clearer! We prefer to keep these on the same axis to order tasks by the gap and show the lack of correlation between bars. You’re also right that the x-axis lists discrete model variants in Fig. 2. It wouldn’t be appropriate to interpolate along the lines and we hoped the axis labels made that clear. Our intention was to group models by family and order these by their underlying latent variables (model size and reasoning capability). The connecting lines are intended to reflect trends along these variables. We’d be happy to keep iterating, but we believe that’s appropriate and these are just our thoughts behind the current revision.
>
> - **[Q1]** We agree that this is a very interesting question. In fact, we originally hoped to include experiments to confirm the causal effect. However, we found that it was hard in practice to induce linearity in the embedding space without re-pretraining the model. We attempted to fine-tune the model (updating the whole model or just the embedding matrix; 10,000 steps, varying batch sizes and LRs) on our non-linearly-represented tasks, but were unable to induce any significant change in linearity. The model was likely able to fit well with shallower updates. [1] does find a correlation between co-occurrence frequency and representational linearity, but determining causality would likely require manipulating the training data and pre-training models. For now, we present strong correlational evidence to support the relationship. We are working on new experimental designs to test this, we’d love any specific ideas/insights you might have for how to manipulate this in already trained models!

---

> > ### Author Response · Authors · 2025-11-23
> >
> > - **[Q2]** We intuitively expect a relationship between use of the compositional mechanism and ability to generalize. However, it is difficult to actually measure generalization in pre-trained models, because this assumes knowledge of what is seen vs. unseen in the training data. We generally address connections of our work to compositional generalization in [L388-401]. Our experiment in App. G shows that we can (partially) replace $g$ with some $g’$ during invocation of the compositional mechanism. Future work might conduct similar experiments, controlling for some seen $g(f(x))$ and $g’(f’(x’))$ and unseen $g’(f(x))$, to test for generalization by means of the mechanism.
> >
> >
> > [1] Jack Merullo, Noah A. Smith, Sarah Wiegreffe, Yanai Elazar. On Linear Representations and Pretraining Data Frequency in Language Models. ICLR 2025.

---

> > > ### Comment · Reviewer_KcBP · 2025-11-26
> > >
> > > thanks for the explanations, I maintain my scores.

---

### Author Response · Authors · 2025-12-04
**Summary of reviews & discussion**

Thanks to all of the reviewers in their efforts to consider our paper. Given the length of the discussion (and as suggested by ICLR), we'll summarize here and do our best to be faithful to the discussions. Please see the full discussion for details!

**Strengths:**

- **Motivation:** [zRpK] thought our paper was "well motivated"; [NP7L] considers our research question "of high importance for understanding the inner workings of LLM reasoning". [KcBP] "likes that the paper thoughtfully engages ... and positions findings relative to previous work on compositionality and debates about behaviour vs. representation".
- **Setup:** [KcBP] found our "experimental setup ... very clear" and "examinations ... very well-explored through many different types of relationships". [LbGz] also noted that our "rules appeared to be carefully chosen". [zRpK] mentioned our paper was "well structured". [NP7L] said the same, also mentioning clear presentation and sound application of methods.
- **Findings:** All reviewers found our paper's core findings "very interesting". [zRpK] remarked that "each section reveals distinct and insightful aspects of factual knowledge encoded in LLMs".
  1. [KcBP, zRpK, NP7L] were especially interested in the relationship between the embedding-space mappings and internal mechanisms, "[posing] new insights on LLM reasoning [and] allowing prediction of inspected phenomena beyond inspected models" [NP7L].
  2. [LbGz] found our "distinction between direct and compositional mechanisms ... interesting and [empirically] well supported".
  3. [zRpK, NP7L, LbGz] expressed interest in our findings on the compositionality gap with our new datasets and on modern models. [LbGz] pointed out that the "evidence for a shrinking compositionality gap is novel and very interesting if true" (as Press 2022 otherwise showed a gap that grows with model size), and that it is also interesting that "reasoning models still struggle with compositionality".

**Weaknesses & Questions:** Despite these positive remarks, initial scores trended negatively. Some suggestions were very constructive, but many seemed to stem from misunderstandings about our research question and setup, which we believe contributed to a low perception of our work's "soundness". We maintain belief in our goals and experimental designs. We did our best to substantively respond to all points and re-clarify as needed.

- Several reviewers wondered whether our findings generalized to additional models [KcBP, zRpK, NP7L]. We did too and we repeated all main body experiments on OLMo 2 (7B & 13B) and Llama 3 (3B) Instruct. Our new results equally or even more strongly support our findings and hypothesis: e.g. our linearity–mechanism relationship continues to strongly correlate in these models ($r^2 = 0.75, 0.50, 0.63$ respectively).

- We provided [KcBP] several practical implications of our work.

Interesting experiments we consider out of scope:

- [KcBP] and [zRpK] were interested in causal evidence for our hypothesized relationship. For now, we present strong and consistent correlations. We previously attempted to make targeted causal interventions, but faced difficulty in manipulating pre-trained models. We're working towards more intensive efforts to measure causality: e.g. by pre-training models from scratch.

- [NP7L] and [LbGz] were interested in analyses of reasoning models. But it's unclear whether reasoning traces are faithful and otherwise difficult to apply current interpretability methods to these models.

- [KcBP] wanted to know how our work relates to compositional generalization. We discuss this in [L388-401], but it is hard to actually measure generalization in pre-trained models. We believe this is suitable for future work.

- [zRpK] was interested in the effect of chain-of-thought prompting and performance on other prompts. We emphasize that we tested reasoning models in Sec. 3 to address the same underlying motivation. We tested more prompt variants to demonstrate robustness.

Points we clarified / disagreed with:
- [KcBP] claims we discard multi-token examples in our analyses.
- [NP7L] believed we were missing a failure analysis, and details on our prompting and methods.
- We reminded [NP7L] that our reasoning budget was 2000 tokens, which should be generous for 2-hop tasks.
- [LbGz] was concerned with our simple method for representing input embeddings. We do this to directly model embedding spaces and find strong correlations regardless.
- [LbGz] and [NP7L] believed our compositional task was disproportionately harder due to the need to infer functions from examples. We clarified our task setup and how it is specifically designed to measure the compositionality gap. We conducted additional experiments to show the effects of prompting with task instructions.

Minor points: We adjusted Fig. 1 based on [KcBP]'s feedback; we provided example model predictions to [NP7L] for validating our evaluation; we will cite the papers mentioned by [LbGz].

---

### Meta-Review · Area_Chair_wzL3 · 2026-01-05

**Summary:**

Reviewers agree the question is well-motivated, and the work is potentially novel, but raised concerns about (i) limited scope (initially focused on a single small model / narrow prompting regime), (ii) confounds between “compositional reasoning” and in-context rule induction from examples, and (iii) the evidentiary gap between observed linear-geometry effects and causal mechanistic claims.

The rebuttal adds experiments on additional model families (e.g., OLMo 2 and Llama 3 instruct) and clarifies methodology; at least one skeptical reviewer explicitly raised their score. Nevertheless, multiple reviewers’ core reservations (causality and task-design confounds) remain only partially resolved, supporting a borderline-reject meta-decision.

**Reviewer Concerns:**

Addressed by rebuttal:

•	Generality across models: added experiments beyond the original setting (OLMo 2 7B/13B; Llama 3 instruct), directly responding to scope/generalization critiques.

•	Method/prompting clarity: One reviewer’s detailed methodological concerns were addressed mainly to the point of a score increase.

Still outstanding:

•	Causality vs correlation: reviewers asked whether the linear geometry is causally tied to the mechanism; this remains unresolved, mainly and framed as difficult.

•	Task-design confound (composition vs rule induction): reviewers note the possibility that inferring rules from few-shot examples drives results; rebuttal does not eliminate this alternative explanation.

•	Prompting/output constraints and ecological validity: concerns remain about the prompting setup and evaluation constraints (e.g., older prompting conventions; limitations to single-token outputs).

**Reviewer Scores:**

zRpK: initial 6, likely stays 6 given added cross-model experiments, though causal concerns persist.

KcBP: intial 4, explicitly indicates their assessment would not change after the authors’ updates; stays 4.

NP7L: initial 2, raise to 4.

LbGz: initial 4, given remaining causal/task-design doubts, likely stays 4.

---

### Decision · Program_Chairs · 2026-01-26

Reject